# Segment Anything without Supervision

**XuDong Wang    Jingfeng Yang    Trevor Darrell**

UC Berkeley

code: https://github.com/frank-xwang/UnSAM

## Abstract

The Segmentation Anything Model (SAM) requires labor-intensive data labeling. We present Unsupervised SAM (UnSAM) for promptable and automatic whole-image segmentation that does not require human annotations. UnSAM utilizes a divide-and-conquer strategy to "discover" the hierarchical structure of visual scenes. We first leverage top-down clustering methods to partition an unlabeled image into instance/semantic level segments. For all pixels within a segment, a bottom-up clustering method is employed to iteratively merge them into larger groups, thereby forming a hierarchical structure. These unsupervised multi-granular masks are then utilized to supervise model training. Evaluated across seven popular datasets, UnSAM achieves competitive results with the supervised counterpart SAM, and surpasses the previous state-of-the-art in unsupervised segmentation by 11% in terms of AR. Moreover, we show that supervised SAM can also benefit from our self-supervised labels. By integrating our unsupervised pseudo masks into SA-1B's ground-truth masks and training UnSAM with only 1% of SA-1B, a lightly semi-supervised UnSAM can often segment entities overlooked by supervised SAM, exceeding SAM's AR by over 6.7% and AP by 3.9% on SA-1B.

## 1   Introduction

Trained on massive unlabeled data using self-supervised learning methods, Large Language Models (LLMs) [5, 34, 33, 46, 2, 19] in natural language processing have revolutionized our world and redefined human-computer interactions. In the domain of computer vision, the recent introduction of the Segment Anything Model (SAM) [21] has dramatically transformed the field with its exceptional ability to handle diverse image segmentation tasks. However, the need for comprehensive manual labeling of training data—over 20 minutes per image [21]—limits SAM from following the scaling laws that benefit LLMs [20]. As a result, despite SA-1B [21] being the most extensive segmentation dataset available, it contains only about 11 million images. Moreover, human-annotated data often introduces significant biases based on the annotators' perceptions of "what constitutes an instance", which frequently leads to the oversight of small entities within the images.

This challenge raises a crucial question addressed in this paper: *Can we "segment anything" without supervision?* In response, we present **UnSAM**, an innovative unsupervised learning method capable of performing both interactive and whole-image segmentation without the need for supervision.

How can we achieve fine-grained and multi-granular segmentation masks comparable to those in SA-1B [21] without supervision? Insights from neuroscience suggest that the human visual system exploits the structure of visual scenes by decomposing dynamic scenes into simpler parts or motions. This perception of hierarchically organized structures implies a powerful "divide-and-conquer" strategy for parsing complex scenes [4, 27]. Drawing inspiration from this, we introduce a divide-and-conquer approach designed to generate hierarchical image segmentation results directly from raw, unlabeled images. The divide-and-conquer approach is a crucial element of UnSAM, enabling it to effectively parse and segment images at multiple levels of granularity.

38th Conference on Neural Information Processing Systems (NeurIPS 2024).

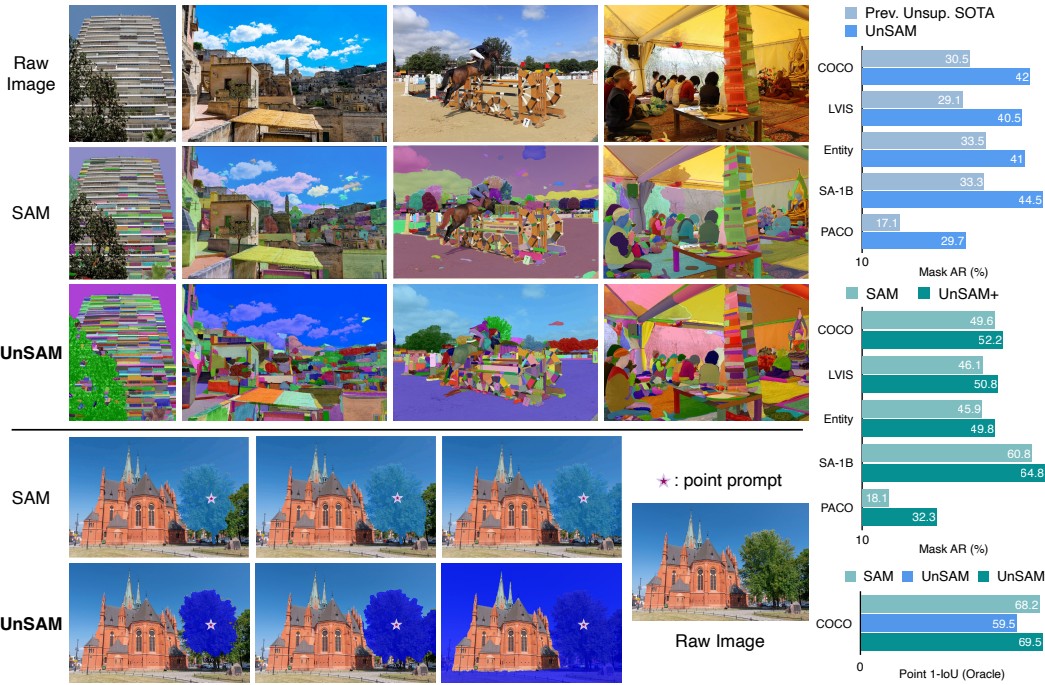

**Figure 1:** UnSAM significantly surpasses the performance of the previous SOTA methods in unsupervised segmentation, and delivers impressive whole image and promptable segmentation results, rivaling the performance of the supervised SAM [21]. This comparative analysis features our unsupervised UnSAM, the supervised SAM, and an enhanced version, UnSAM+, across a variety of datasets. The top section displays raw images (row 1) alongside whole image segmentation outputs from UnSAM (row 3), and SAM (row 2). The bottom section highlights our promptable segmentation results using a point prompt (*i.e.*, the star mark). The right panel quantitatively compares the performance across models, including metrics like Mask AR (%) and Point IoU.

Our pseudo-mask generation pipeline initiates with a top-down clustering approach (*i.e.*, the divide stage), to extract initial semantic and instance-level masks using a Normalized Cuts-based method CutLER [39, 31]. Subsequently, UnSAM refines these masks using a bottom-up clustering method (*i.e.*, the conquer stage): within each mask, we iteratively merge semantically similar pixels into larger segments based on various similarity thresholds. The resulting masks at different thresholds in the conquer stage, along with the masks produced in the divide stage, create a hierarchical structure. Technically, we can generate a vast range of granularities with minimal extra cost! Furthermore, UnSAM captures more subtle details that pose challenges for human annotators, significantly enriching the granularity and utility of unsupervised segmentation models.

Equipped with these sophisticated multi-granular pseudo masks as "ground-truth" labels, UnSAM is adeptly trained to perform both interactive and automatic whole-image segmentation, demonstrating remarkable versatility across various segmentation scenarios. We have observed that our UnSAM model frequently identifies objects that SAM [21] overlooks, particularly types of objects or parts typically missed by ground-truth annotations of SA-1B [21], such as human ears, animal tails, *etc*.

The capabilities of UnSAM are rigorously tested across seven major whole-entity and part segmentation datasets, *e.g.*, MSCOCO [24], LVIS [15], SA-1B [21], ADE [48], Entity [29], PartImageNet [16] and PACO [30]. As illustrated in Fig. 1, we demonstrate some noteworthy behaviors:

- The performance gap between unsupervised segmentation models and SAM can be significantly reduced: By training on just 1% of SA-1B's unlabeled images with a ResNet50 backbone, UnSAM not only advances the state-of-the-art in unsupervised segmentation by 10% but also achieves comparable performance with the labor-intensive, fully-supervised SAM.

- The supervised SAM can also benefit from our self-supervised labels: integrating our unsupervised pseudo masks with SA-1B's ground-truth data and retraining UnSAM on this combined data enables UnSAM+ to outperform SAM's AR by over 6.7% and AP by 3.9%. We observed that UnSAM and UnSAM+ can often discover entities missed by SAM.

## 2 Related Works

### 2.1 Self-supervised Image Segmentation

Recent advances in unsupervised image segmentation [39, 28, 44, 6, 8, 41, 38, 42, 35, 12, 7, 37] have leveraged the emergent segmentation capabilities of self-supervised Vision Transformers (ViT) [8, 14, 17] to "discover" objects within images. Initial efforts, such as TokenCut [44] and LOST [32], have produced semantically meaningful pixel groupings for salient objects by utilizing the class-attention mechanism of self-supervised ViTs. As a representative work in the unsupervised segmentation domain, CutLER [39] introduced a cut-and-learn pipeline for unsupervised object detection and image segmentation. CutLER initially generates high-quality pseudo masks for multiple objects using MaskCut [39], followed by learning a detector on these masks using a loss dropping strategy. Extending this approach, VideoCutLER [40] employs a cut-synthesis-and-learn strategy for segmenting and tracking multiple instances across video frames without supervision. Additionally, SOHES [6] introduced the global-local self-exploration method to cluster image features from high to low cosine similarity, obtaining pseudo masks that cover multiple hierarchical levels.

In contrast, UnSAM introduces a divide-and-conquer pipeline that generates more pseudo masks per image at the same processing speed, but with enhanced quality and broader coverage across hierarchical levels. Furthermore, UnSAM captures more subtle details that pose challenges for human annotators, significantly enriching the granularity and utility of unsupervised segmentation models.

### 2.2 Promptable Image Segmentation

Tradition segmentation models have focused on predicting masks for all instances or semantic parts within a single image simultaneously. Recently, however, models have begun to interact with users, generating segmentation masks based on user inputs such as points [21, 23, 47, 45, 11], text descriptions [26], or bounding boxes [21]. Moreover, some approaches now frame segmentation tasks within an in-context learning framework [43, 3], utilizing in-context examples to define distinct segmentation tasks. For example, the Segment Anything model [21] can produce masks in a zero-shot manner based on different types of prompts. One limitation of SAM is that it only produces three class-agnostic masks. An extension, Semantic-SAM [23], aims to segment and recognize objects at multiple granularities through a multi-choice learning scheme, allowing each click point to produce masks at multiple levels along with their semantic labels. Nevertheless, both models are supervised and rely on large-scale, human-annotated data, which introduces issues of annotator bias and scalability limitations.

In contrast, our unsupervised UnSAM and lightly semi-supervised UnSAM+ model demonstrate superior performance in the promptable segmentation task, offering a robust alternative to these fully-supervised approaches.

## 3 Preliminaries

### 3.1 Cut and Learn (CutLER) and MaskCut

CutLER [39] introduces a cut-and-learn pipeline to precisely segment instances without supervision. The initial phase, known as the cut stage, uses a normalized cut-based method, MaskCut [39], to generate high-quality instance masks given the patch-wise cosine similarity matrix $W_{ij} = \frac{K_i K_j}{|K_i|_2 |K_j|_2}$, where $K_i$ is "key" features of patch $i$ in the last attention layer of unsupervised ViT. To extract multiple instance masks from a single image, MaskCut repeats this operation but adjusts by masking out patches from previously segmented instances in the affinity matrix: $W_{ij}^t = \frac{\left(K_i \sum_{s=1}^t M_{ij}^s\right)\left(K_j \sum_{s=1}^t M_{ij}^s\right)}{\|K_i\|_2 \|K_j\|_2}$ Subsequently, CutLER's learning stage trains a segmentation/detection model on these pseudo-masks with drop-loss. Please check Appendix A.2 for more details on CutLER.

### 3.2 Segment Anything Model (SAM) and SA-1B

Segment Anything [21] tackles the promptable segmentation task. At its core lies the Segment Anything Model (SAM), which is capable of producing segmentation masks given user-provided points, boxes, and masks in a zero-shot manner. One significant contribution of SAM is the release of

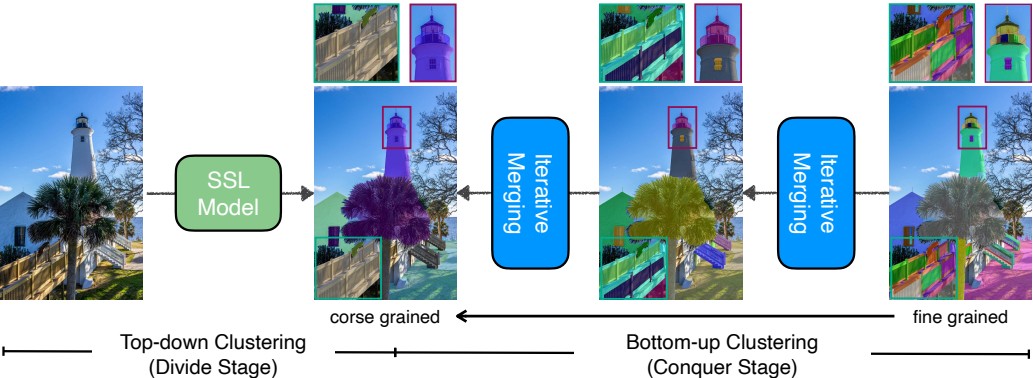

Top-down Clustering
(Divide Stage)

corse grained

Bottom-up Clustering
(Conquer Stage)

fine grained

**Figure 2:** Our divide-and-conquer pipeline for generating the "ground-truth" pseudo masks used for training UnSAM without human supervision begins with a top-down clustering approach (*i.e.*, the divide stage), to extract initial semantic/instance-level masks using a Normalized Cuts [31]-based CutLER [39]. Subsequently, we refine these masks using a bottom-up clustering method (*i.e.*, the conquer stage): within each mask, we iteratively merge semantically similar pixels into larger segments using various similarity thresholds. The resulting masks at different thresholds create a hierarchy. We zoom-in selected regions to visualize details.

the SA-1B dataset [21], which comprises 11M high-resolution images and 1.1 billion segmentation masks, providing a substantial resource for training and evaluating segmentation models. While SAM significantly accelerates the labeling of segmentation masks, annotating an image still requires approximately 14 seconds per mask. Given that each image contains over 100 masks, this equates to more than 30 minutes per image, posing a substantial cost and making it challenging to scale up the training data effectively. For more details on SAM and SA-1B, please check Appendix A.3.

## 4 UnSAM: Segment Anything without Supervision

### 4.1 Divide-and-Conquer for Hierarchical Image Segmentation

Our segment anything without supervision model starts by generating pseudo masks that respect the hierarchical structure of visual scenes without supervision. This approach is motivated by the observation that the "divide and conquer" strategy is a fundamental organizational principle employed by the human visual system to efficiently process and analyze the vast complexity of visual information present in natural scenes [4, 27]. Our pseudo-mask generation pipeline **divide-and-conquer**, which is summarized in Alg. 1 and illustrated in Fig. 2, consists of two stages:

**Divide stage**: we leverage a Normalized Cuts (NCuts)-based method, CutLER [39, 31], to obtain semantic and instance-level masks from unlabeled raw images. CutLER's cut-and-learn pipeline and its MaskCut method are discussed in Sec. 3.1. However, the coarser-granularity masks predicted by CutLER can be noisy. To mitigate this, we filter out masks with a confidence score below a threshold $\tau$. Empirically, salient semantic and instance-level entities typically encompass richer part-level entities (for example, a person has identifiable parts such as legs, arms, and head, whereas a background sky contains few or no sub-level entities). To extract these part-level entities with a hierarchical structure, we employ a conquer phase.

**Conquer stage**: for each instance-/semantic-level mask discovered in the previous stage, we employ iterative merging [1, 6] to decompose the coarse-grained mask into simpler parts, forming a hierarchical structure.

More specifically, we first crop local patches using the masks we obtained in the divide phase, and bi-linearly interpolate local patches to the resolution of $256 \times 256$. We then feed them into DINO pre-trained ViT-B/8 [8] encoder $f(\cdot)$, and extract 'key' features $k_i = f(p_i)$ from the last attention layer as patch-wise features for local patches $p_i$. Subsequently, the conquer phase employs iterative merging [1, 6] to group patches into larger clusters, with pre-defined cosine similarity thresholds at $\theta \in \{\theta_1, ..., \theta_l\}$, where $l$ is the predefined granularity levels.

---

**Algorithm 1** Divide and Conquer

---

$I_{\text{resized}} \leftarrow$ input image $I$ resized to $1024 \times 1024$
$M \leftarrow \{m : m \in \text{CutLER}(I_{\text{resized}}) \wedge m_{\text{score}} > \tau\}$
**for** $m \in M$ **do**
    Add $m$ into $S_0$
    $bbox \leftarrow$ bounding box $[x_1, y_1, x_2, y_2]$ of $m$
    $I_{\text{local}} \leftarrow I_{\text{resized}}$ cropped by $bbox$, resized to $256 \times 256$
    $K \leftarrow \text{DINO}(I_{\text{local}})$
    **for** $\theta_t \in \theta_l, \dots, \theta_1$ **do**
        **if** $t = l$ **then**
            Initialize $k_i^t \leftarrow K_i, C_i^t \leftarrow p_i \ \forall i, a \leftarrow 1$
            where $p_i$ is corresponding patch of $K_i$, add $p_i$ into $S_l \ \forall i$
        **else**
            Initialize $S_t \leftarrow S_{t+1}, k_i^t \leftarrow k_i^{t+1}, C_i^t \leftarrow C_i^{t+1} \ \forall i$
        **end if**
        **while** $a \geq \theta_t$ **do**
            Identify adjacent $p_i, p_j$ with $i, j \leftarrow \underset{i,j}{\text{argmax}} \frac{k_i^t k_j^t}{||k_i^t||_2 ||k_j^t||_2}, a \leftarrow \underset{i,j}{\max} \frac{k_i^t k_j^t}{||k_i^t||_2 ||k_j^t||_2}$
            Identify cluster $C_m^t, C_n^t$, where $p_i \in C_m^t, p_j \in C_n^t$
            Remove $C_m^t$ and $C_n^t$ from $S_t$
            $C^t \leftarrow C_m^t \cup C_n^t$, add $C^t$ into $S_t$
            $\forall p_z \in C^t, k_z^t \leftarrow \frac{a_m k_i^t + a_n k_j^t}{a_m + a_n}$, where $a_m$ is the size of cluster $C_m^t$ and $p_i \in C_m^t$
        **end while**
    **end for**
**end for**

---

In iteration $t$, our method finds two adjacent patches $(p_i, p_j)$ from two separate clusters $(C_m^t, C_n^t)$ with the highest cosine similarity $\frac{k_i^t k_j^t}{||k_i^t||_2 ||k_j^t||_2}$, merges them into one cluster, and updates $k_i^t$ and $k_j^t$ to $\frac{a_m k_i^t + a_n k_j^t}{a_m + a_n}$, where $a_m$ is the number of patches in cluster $C_m^t (p_i \in C_m^t)$. The conquer stage repeats this step until the maximum cosine similarity is less than $\theta_t$, collects all merged clusters as new part-level pseudo masks, and uses smaller threshold $\theta_{t+1}$ to iterate again. Each coarse-grained mask discovered in the divide stage can form a hierarchical structure $H$ after the conquer stage:

$$H = \{S_0, S_1, ..., S_t, ..., S_l\}, \text{where } S_t = \{C_1^t, ..., C_{n_t}^t\}, n_i \leq n_j \text{ if } i < j \tag{1}$$

$n_t$ is the number of clusters/masks belonging to granularity level $t$ and $n_0 = 1$.

**Mask merging:** The new part-level pseudo masks discovered in the conquer stage are added back to the semantic and instance-level masks identified in the divide stage. We then use Non-Maximum Suppression (NMS) to eliminate duplicates. Following previous works in unsupervised image segmentation [39, 28, 6], we also employ off-the-shelf mask refinement methods, such as Conditional Random Fields (CRF) [22] and CascadePSP [10], to further refine the edges of the pseudo masks. Finally, we filter out the post-processed masks that exhibit significant differences in Intersection-over-Union (IoU) before and after refinement.

**Preliminary results:** The divide-and-conquer pipeline achieves a pseudo mask pool with more entities, a broader range of granularity levels, and superior quality compared to previous work, *e.g.*, CutLER [39], U2Seg [28] and SOHES [6]. As shown in Table 3, its pseudo masks reach 23.9% AR on 1000 randomly selected validation images from the SA-1B dataset [21], representing a 45.7% improvement over the state-of-the-art.

**Key distinctions over prior works on pseudo-mask generation:** The divide-and-conquer strategy employed by UnSAM sets it apart from previous works:

[39, 28] rely solely on top-down clustering methods, providing only instance and semantic-level masks, and thereby missing the hierarchical structure present in complex images. In contrast, our pipeline captures this hierarchical structure by identifying more fine-grained pixel clusters.

While [6] does incorporate some hierarchical structure through bottom-up clustering with iterative merging, it still misses many fine-grained instances and some large-scale instance masks. Additionally, the iterative merging in [6] focuses on small regions below a certain mask size threshold, primarily to refine noisy small masks, limiting its ability to detect a full range of entity sizes. Our experimental results demonstrate qualitatively and quantitatively superior performance compared to prior works, particularly in producing high-quality, detailed pseudo-masks that better capture the hierarchical complexity of visual scenes.

## 4.2 Model Learning and Self-Training

Although the pseudo masks generated by our pipeline are qualitatively and quantitatively superior to those from prior works, they can still be somewhat noisy. Our self-supervised pipeline has limitations in identifying certain types of instances. For example, iterative merging sometimes fails to correctly associate disconnected parts of the same entity. To address this, we utilize a self-training strategy to further enhance UnSAM's model performance. UnSAM learns an image segmentation model using the masks discovered by the divide-and-conquer strategy. It has been observed that self-training enables the model to "clean" the pseudo masks and predict masks of higher quality [39]. Once we have prepared the pseudo-masks, UnSAM can be integrated with any arbitrary whole-image or promptable image segmentation models during the model learning or self-training stage.

**Whole-image segmentation**. We choose the vanilla Masked Attention Mask Transformer (Mask2Former) [9] for simplicity. The key innovation of Mask2Former is the introduction of a masked attention mechanism in the transformer's cross-attention block, defined as $\text{softmax}(M + QK^T)V$, where the attention mask $M$ at feature location $(x, y)$ is given by: $M(x, y) = \begin{cases} 0 & \text{if } M(x, y) = 1 \\ -\infty & \text{otherwise} \end{cases}$. This mechanism constrains attention within the region of the predicted mask. UnSAM is then trained using the following mask prediction loss:

$$\mathcal{L} = \lambda_{\text{ce}}\mathcal{L}_{\text{ce}} + \lambda_{\text{dice}}\mathcal{L}_{\text{dice}} \tag{2}$$

where $\mathcal{L}_{\text{ce}}$ and $\mathcal{L}_{\text{dice}}$ is the cross-entropy and Dice loss, with $\lambda_{\text{ce}}$ and $\lambda_{\text{dice}}$ as their respective weights.

After one round of self-training UnSAM on the pseudo-masks, we perform a second round of self-training by merging high-confidence mask predictions (with a confidence score greater than $\tau_{\text{self-train}}$) as the new 'ground-truth' annotations. To avoid duplication, we filter out ground truth masks that have an IoU greater than 0.5 with the predicted masks.

**Promptable Image Segmentation**. Similar to SAM [21], our unsupervised SAM can also produce high-quality object masks from input prompts such as points. We utilize Semantic-SAM [23] as the base model for predicting multiple granularity levels of masks from a single click. During the learning process, we randomly sample points within an inner circle (radius $\leq 0.1 \cdot \min(\text{Mask}_{\text{width}}, \text{Mask}_{\text{height}})$) of the mask to simulate user clicks.

## 4.3 UnSAM+: Improving Supervised SAM with Unsupervised Segmentation

The supervised SAM model's [21] reliance on human-annotated data introduces a significant bias based on the annotator's perception of *'what constitutes an instance'*, frequently missing some entities within the image. In contrast, since our mask generation pipeline does not rely on human supervision, it can often identify valid objects or parts that are overlooked by SA-1B's [21] ground-truth annotations.

Motivated by this observation, we leverage UnSAM to improve the performance of the supervised SAM [21] by implementing a straightforward yet effective strategy: merging SA-1B's ground-truth masks $D_{\text{SA-1B}}$ with our unsupervised segmentation masks $D_{\text{UnSAM}}$ based on the IoU, formulated as:

$$D^i_{\text{UnSAM+}} = D^i_{\text{SA-1B}} \cup \{\forall C_m \in D^i_{\text{UnSAM}} \text{ if } \text{IoU}^{\text{max}}(C_m, \forall C_n \in D^i_{\text{SA-1B}}) \leq \tau_{\text{UnSAM+}}\} \tag{3}$$

$\tau_{\text{UnSAM+}}$ is the IoU threshold, $\text{IoU}^{\text{max}}$ is the maximum IoU between $C_m$ and any mask $C_n$ in $D^i_{\text{SA-1B}}$, and $D^i_{\text{SA-1B}}$ and $D^i_{\text{UnSAM+}}$ is the set of SA-1B and unsupervised masks within image $i$, respectively. We then train UnSAM+ on $D_{\text{UnSAM+}}$ for promptable image segmentation and whole-image segmentation. The fusion approach leverages the strengths of both supervised and unsupervised annotations, addressing the limitations inherent in human-annotated datasets while significantly enriching the diversity and comprehensiveness of the training data. This results in a more robust and generalizable segmentation model UnSAM+, surpassing the performance of SAM.

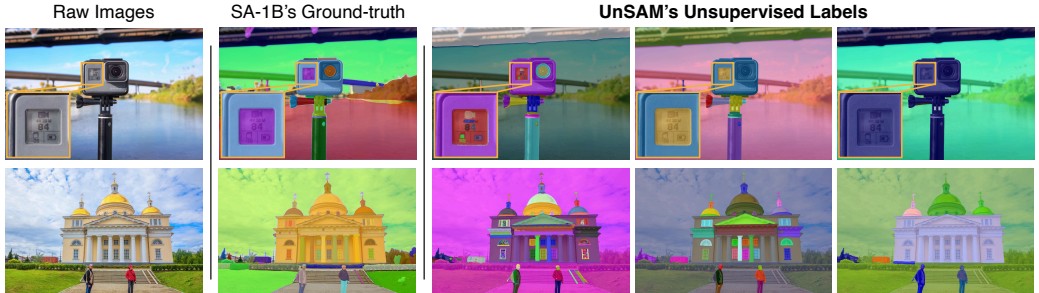

**Figure 3:** Unsupervised pseudo-masks generated by our divide-and-conquer pipeline not only contain precise masks for coarse-grained instances (column 5), *e.g.*, cameras and persons, but also capture fine-grained parts (column 3), *e.g.*, digits and icons on a tiny camera monitor that are missed by SA-1B's [21] ground-truth labels.

| Methods | Backbone (# params) | # images | Avg. | Datasets with Whole Entities | | | | | Datasets w/ Parts | |
|---|---|---|---|---|---|---|---|---|---|---|
| | | | | COCO | LVIS | ADE | Entity | SA-1B | PtIn | PACO |
| SAM (supervised) | ViT-B/8 (85M) | 11M | 42.1 | 49.6 | 46.1 | 45.8 | 45.9 | 60.8 | 28.3 | 18.1 |
| FreeSOLO [41] | RN-101 (45M) | 1.3M | 7.3 | 11.6 | 5.9 | 7.3 | 8.0 | 2.2 | 13.8 | 2.4 |
| CutLER [39] | RN-50 (23M) | 1.3M | 21.8 | 28.1 | 20.2 | 26.3 | 23.1 | 17.0 | 28.7 | 8.9 |
| SOHES [6] | ViT-B/8 (85M) | 0.2M | 30.1 | 30.5 | 29.1 | 31.1 | 33.5 | 33.3 | 36.0 | 17.1 |
| UnSAM | RN-50 (23M) | 0.1M | **39.2** | **40.5** | **37.7** | **35.7** | **39.6** | **41.9** | **51.6** | **27.5** |
| UnSAM | RN-50 (23M) | 0.2M | **40.4** | **41.2** | **39.7** | **36.8** | **40.3** | **43.6** | **52.1** | **29.1** |
| UnSAM | RN-50 (23M) | 0.4M | **41.1** | **42.0** | **40.5** | **37.5** | **41.0** | **44.5** | **52.7** | **29.7** |
| *vs. prev. SOTA* | | | **+11.0** | **+11.5** | **+11.4** | **+6.4** | **+7.5** | **+11.2** | **+16.7** | **+12.6** |

**Table 1:** UnSAM achieves the state-of-the-art results on unsupervised image segmentation, using a backbone of ResNet50 and training with only 1% of SA-1B [21] data. We perform a zero-shot evaluation on various image segmentation benchmarks, including whole entity datasets, *e.g.*, COCO and ADE, and part segmentation datasets, *e.g.*, PACO and PartImageNet. The evaluation metric is average recall (AR).

## 5 Experiments

### 5.1 Model Training Settings

We provide a brief overview of the model training settings and include more details in Appendix A.1.

**Pseudo mask generation.** In the divide stage, we set the confidence threshold $\tau$=0.3; in the conquer stage, we choose threshold $\theta_{merge} = [0.6, 0.5, 0.4, 0.3, 0.2, 0.1]$. When merging the pseudo masks with the ground truths for training UnSAM+, we select $\tau_{\text{UnSAM+}} = 0.02$. **Whole-image segmentation.** UnSAM picks DINO [8] pre-trained ResNet-50 [18] as the backbone and Mask2former [9] as the mask decoder. The default learning rate is $5 \times 10^{-5}$ with a batch size of 16 and a weight decay of $5 \times 10^{-2}$. We train the model for 8 epochs. **Promptable segmentation.** UnSAM uses the self-supervised pre-trained Swin-Transformer [25] Tiny model as the backbone, and leverages Semantic-SAM [23] as the base model. We set the number of hierarchy levels to 6, which is also the number of predicted masks UnSAM generates per prompt during inference. One can easily train with a different number of granularity levels as needed. For all experiments, we train UnSAM with 1~4% unlabeled images from SA-1B dataset [21].

### 5.2 Evaluation Datasets and Metrics

**Whole-image segmentation.** To evaluate our model's performance, we test our models on various datasets in a zero-shot manner to evaluate the performance of segmenting entities from all granularity levels. We choose COCO [24], LVIS [15], ADE20K [48], EntitySeg [29], and SA-1B [21] that mainly encompass semantic-/instance-level entities; PartImageNet [16] and PACO [30] that cover part-level entities. The SA-1B test set consists of randomly selected 1000 images not included in our training set. Notably, each dataset only covers entities from certain hierarchical levels and certain pre-defined classes, while our model generates masks from all levels and all classes. Hence, the COCO Average Precision (AP) metric could not reflect our model's authentic performance in segmenting all entities in the open-world. Following prior work [39, 6], we mainly consider Average Recall (AR) to compare with different models.

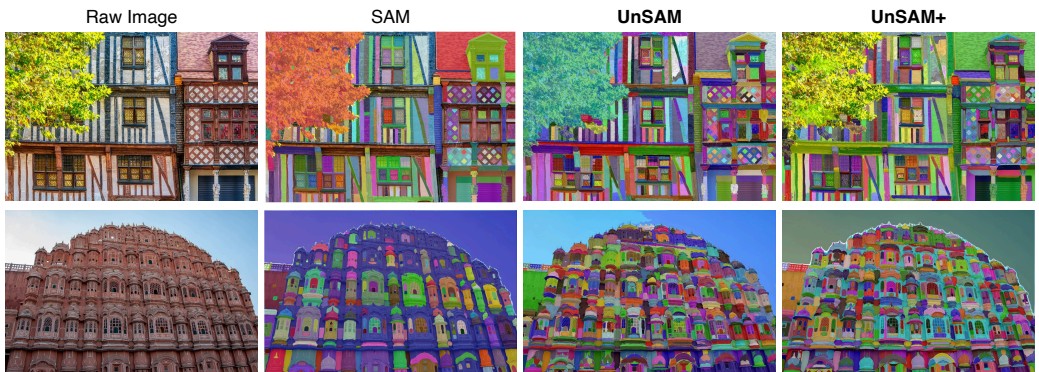

| Raw Image | SAM | **UnSAM** | **UnSAM+** |

**Figure 4:** UnSAM has competitive dense object segmentation results compared to the supervised SAM [21].

| Methods | Backbone (# params) | Sup. Labels | Unsup. Labels | # images | Avg. | Datasets with Whole Entities | | | | | Datasets w/ Parts | |
|---|---|---|---|---|---|---|---|---|---|---|---|---|
| | | | | | | COCO | LVIS | ADE | Entity | SA-1B | PtIn | PACO |
| SAM | ViT-B/8 (85M) | ✓ | ✗ | 11M | 42.1 | 49.6 | 46.1 | **45.8** | 45.9 | 60.8 | 28.3 | 18.1 |
| UnSAM | RN-50 (23M) | ✗ | ✓ | 0.1M | 39.2 | 40.5 | 37.7 | 35.7 | 39.6 | 41.9 | **51.6** | 27.5 |
| UnSAM+ | RN-50 (23M) | ✓ | ✓ | 0.1M | **48.8** | **52.2** | **50.8** | 45.3 | **49.8** | **64.8** | 46.0 | **32.3** |
| *vs. SAM* | | | | | **+6.7** | **+2.6** | **+4.7** | -0.5 | **+3.9** | **+4.0** | **+17.7** | **+14.2** |

**Table 2:** UnSAM+ can outperform SAM [21] on most experimented benchmarks (including SA-1B [21]), when training UnSAM on 1% of SA-1B with both ground truth masks and our unsupervised labels. This demonstrates that our unsupervised pseudo masks can serve as a powerful add-on to the densely annotated SA-1B masks!

| Methods | $AR_{1000}$ | $AR_S$ | $AR_M$ | $AR_L$ |
|---|---|---|---|---|
| SOHES (CRF [22]) | 12.0 | 3.5 | 9.5 | 20.7 |
| SOHES (CascadePSP [10]) | 16.4 | 6.0 | 15.8 | 22.6 |
| UnSAM (CRF [22]) | 15.3 | 2.3 | 11.9 | 27.7 |
| UnSAM (CascadePSP [10]) | 23.9 | 7.9 | 22.4 | 34.0 |
| *vs. prev. SOTA* | **+7.5** | **+1.9** | **+6.6** | **+11.4** |

**Table 3:** Evaluation on unsupervised pseudo masks using SA-1B's [21] ground-truth annotations.

| Methods | AP | $AR_S$ | $AR_M$ | $AR_L$ | $AR_{1000}$ |
|---|---|---|---|---|---|
| SAM | 38.9 | 20.0 | 59.9 | **82.8** | 60.8 |
| UnSAM+ | **42.8** | **36.2** | **65.9** | 76.5 | **64.8** |
| *vs. sup. SAM* | **+3.9** | **+16.2** | **+6.0** | -6.3 | **+4.0** |

**Table 4:** Quantitative comparisons between our lightly semi-supervised SAM, UnSAM+, and the fully-supervised SAM [21] on SA-1B [21].

**Point-based promptable segmentation.** We evaluate our point-based interactive segmentation model on MSCOCO Val2017 [24]. Following the previous work on promptable image segmentation [21, 23], we pick two metrics for model evaluation MaxIoU and OracleIoU. For each point prompt, UnSAM predicts 6 masks representing different granularity levels. MaxIoU calculates the IoU between the mask with the highest confidence score among 6 masks, whereas OracleIoU picks the highest IoU between 6 predicted masks and the ground truth. For each mask in a test image, we select its center as the point prompt.

### 5.3 Evaluation Results

**Unsupervised pseudo-masks.** Unsupervised pseudo-masks generated by our divide-and-conquer pipeline not only contain precise masks for coarse-grained instances, but also capture fine-grained parts that are often missed by SA-1B's [21] ground-truth labels, as shown in Fig. 3.

**Whole-image segmentation.** Remarkably, UnSAM outperforms the previous state-of-the-art methods across **all** evaluation datasets as summarized in Table 1. UnSAM demonstrates superior performance compared to the SOTA method even when trained with only 1% SA-1B training data and a backbone of ResNet-50 with only 23M parameters, while the SOTA utilizes twice training data and a backbone with nearly four times the parameters. This implies that UnSAM is a lightweight, easier to train, and less data-hungry model with better zero-shot performance in segmenting entities in the open-world as shown in Figs. 4 and 5. On average, UnSAM surpasses the previous SOTA by 11.0% in AR. When evaluated on PartImageNet [16] and PACO [30] benchmarks, UnSAM exceeds the SOTA by 16.6% and 12.6 %, respectively.

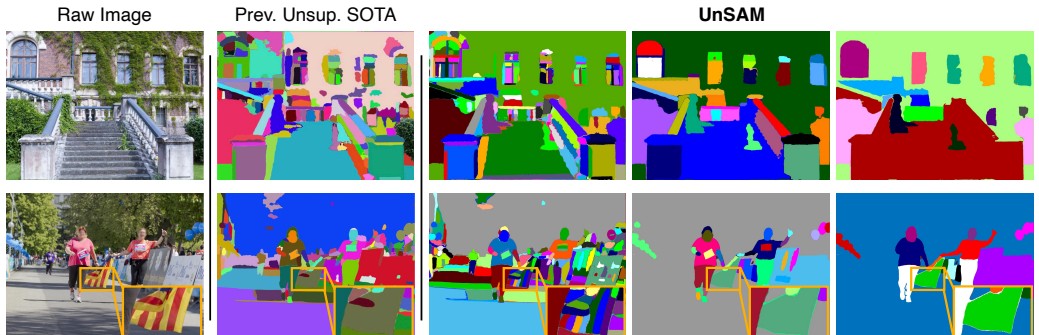

**Figure 5:** UnSAM not only discovers more fine-grained masks than the previous state-of-the-art unsupervised segmentation method [6], but also provides segmentation masks with a wide range of granularity. We show qualitative comparisons between UnSAM (with 3 levels of granularity) and baseline models on SA-1B [21].

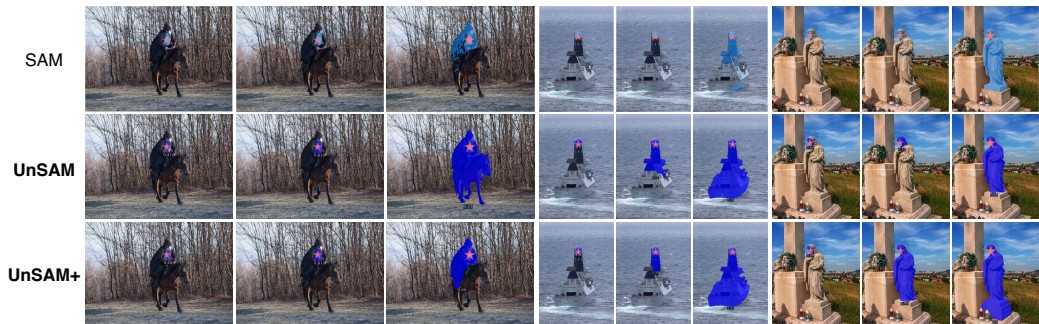

**Figure 6:** Qualitative comparisons of promptable image segmentation between the fully-supervised SAM [21], our unsupervised UnSAM, and the lightly semi-supervised UnSAM+. Both UnSAM and UnSAM+ consistently deliver high-quality, multi-granular segmentation masks in response to the point prompts (*i.e.*, the star mark).

| Methods | Backbone (# params) | Sup. Labels | Unsup. Labels | % of SA-1B | Point (Max) 1-IoU | Point (Oracle) 1-IoU |
|---|---|---|---|---|---|---|
| SAM (B) | ViT-B/8 (85M) | ✓ | ✗ | 100% | 52.1 | 68.2 |
| UnSAM | Swin-Tiny (25M) | ✗ | ✓ | 1% | 40.3 | 59.5 |
| UnSAM+ | Swin-Tiny (25M) | ✓ | ✓ | 1% | **52.4** | **69.5** |

**Table 5:** Despite using a backbone that is 3× smaller and being trained on only 1% of SA-1B, our lightly semi-supervised UnSAM+ surpasses the fully-supervised SAM in promptable segmentation task on COCO.

When compared to the supervised SAM [21], UnSAM's AR across all datasets is already very close, with only a 1% difference. On PartImageNet [16] and PACO [30], UnSAM surpasses SAM by 24.4% and 11.6%. This further demonstrates the excellent capability of our divide-and-conquer pipeline in discovering details that human annotators tend to miss.

Furthermore, our UnSAM+, trained with integrated unsupervised pseudo masks and SA-1B [21] ground truth, outperforms SAM's [21] AR by over 6.7% and AP by 3.9% as shown by Table 2 and 4. UnSAM+ demonstrates superior average recall compared to SAM across all evaluation datasets except for ADE20K [48], which is dominated by semantic-level annotations. UnSAM+'s significantly 16.2 % higher AR on small entities further confirms that our pseudo masks can effectively complement the SA-1B datasets with more details it ignores and the UnSAM+ can often discover entities missed by SAM as demonstrated in Fig. 4 and Fig. 7.

**Point-based promptable segmentation.** As shown in Table 5, UnSAM trained with our pseudo masks achieve 40.3% MaxIoU and 59.5% OracleIoU on COCO. Notably, we train the model with only 1% of the data that SAM [21] uses and a backbone with 4× fewer parameters. Moreover, the UnSAM+ trained with integrated pseudo masks and SA-1B ground truths outperforms SAM on both MaxIoU and OracleIoU with 0.3% and 1.3% respectively. Qualitative results are shown in Fig. 6.

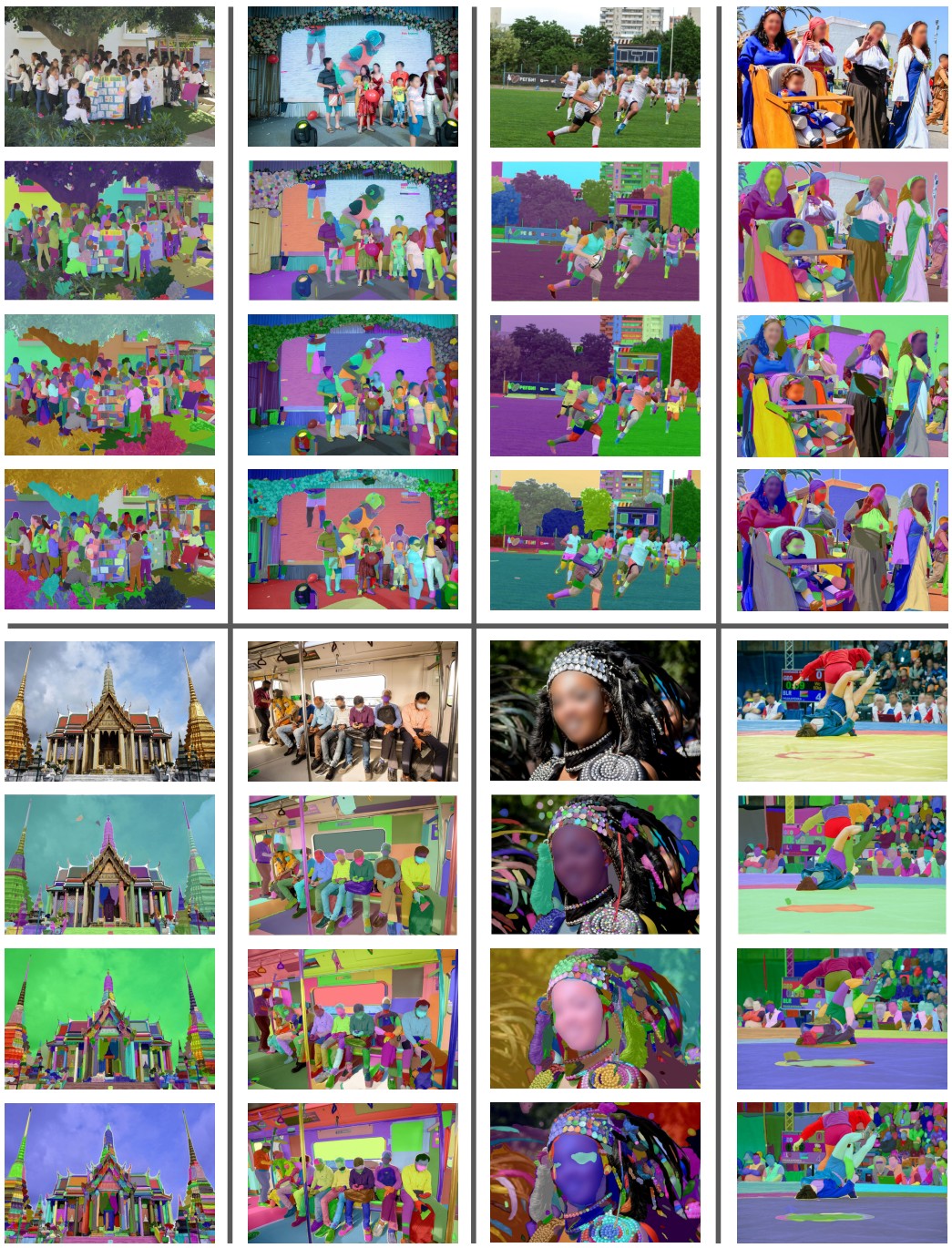

**Figure 7:** More visualizations on SA-1B [21]. From top to bottom are raw images, segmentation by SAM, segmentation by UnSAM, and segmentation by UnSAM+.

## 6  Summary

Image segmentation is a fundamental task in computer vision, traditionally relying on intensive human annotations to achieve a detailed understanding of visual scenes. We propose UnSAM, an unsupervised segmentation model that significantly surpasses the performance of previous state-of-the-art methods in unsupervised image segmentation. Additionally, our unsupervised UnSAM model delivers impressive results, rivaling the performance of the cutting-edge supervised SAM, and exceeding it in certain semi-supervised settings.

**Acknowledgement.** We thank helpful discussions with Jitendra Malik, Cordelia Schmid, Ishan Misra, Xinlei Chen, Xingyi Zhou, Alireza Fathi, Renhao Wang, Stephanie Fu, Qianqian Wang, Baifeng Shi, Max Letian Fu, Tony Long Lian, Songwei Ge, Bowen Cheng and Rohit Girdhar. We thank Shengcao Cao and Hao Zhang for their help in reproducing baseline results. XuDong Wang and Trevor Darrell were funded by DoD including DARPA LwLL and the Berkeley AI Research (BAIR) Commons.

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

# A  Appendix

## A.1  Training Details

**Pseudo mask preparation details.** Empirically, in the divide stage, we set the confidence threshold $\tau = 0.3$; in the conquer stage, we choose threshold $\theta_{merge} = [0.6, 0.5, 0.4, 0.3, 0.2, 0.1]$. For each image, the divide-and-conquer pipeline generates on average 334 pseudo masks. In the self-training phase, the $\tau_{\text{self-train}} = 0.7$, and each image has 448 pseudo masks per image after merging high-confidence mask predictions generated by UnSAM. When merging the pseudo masks with the ground truths for training UnSAM+, we select $\tau_{\text{UnSAM+}} = 0.02$.

**Whole-image segmentation.** UnSAM picks DINO [8] pre-trained ResNet-50 [18] as the backbone and Mask2former [9] as the mask decoder. Given the abundant number of pseudo masks generated, UnSAM augments data only by cropping a $1024 \times 1024$ region from the original image. To cope with a large amount of 'ground-truth' masks per image, we find that having 2000 learnable queries produces the best result. We randomly select at most 200 'ground-truth' masks per image to speed up the training process. The default learning rate is $5 \times 10^{-5}$ with batch size equals 16 and weight decay $5 \times 10^{-2}$. We train the model for 8 epochs. All model training in this paper was conducted using either 4 A100 GPUs or 8 RTX 3090 GPUs.

**Promptable segmentation.** UnSAM uses the self-supervised pre-trained Swin-Transformer [25], specifically the Swin-Tiny model, as the backbone and leverages Semantic-SAM [23] as the base model. Given at most 6 levels of masks corresponding to one input point in SA-1B [21], we set the number of hierarchy levels to 6, which is also the number of predicted masks UnSAM generates per prompt during inference. However, one can easily train with a different number of granularity levels as needed. The default learning rate is $1 \times 10^{-4}$ with a batch size of 8. The learning rate decreases by a factor of 10 at 90% and 95% of the training iterations. We train the model for 4 epochs.

## A.2  Preliminary: Cut and Learn (CutLER) and MaskCut

CutLER [39] introduces a cut-and-learn pipeline to precisely segment instances without supervision. The initial phase, known as the cut stage, uses a normalized cut-based method, MaskCut [39], to generate high-quality instance masks that serve as pseudo-labels for subsequent learning phases. MaskCut begins by harnessing semantic information extracted from "key" features $K_i$ of patch $i$ in the last attention layer of unsupervised vision transformers. It then calculates a patch-wise cosine similarity matrix $W_{ij} = \frac{K_i K_j}{|K_i|_2 |K_j|_2}$. To extract multiple instance masks from a single image, MaskCut initially applies Normalized Cuts [31], which identify the eigenvector $x$ corresponding to the second smallest eigenvalue. The vector $x$ is then bi-partitioned to extract the foreground instance mask $M^s$. Subsequent iterations repeat this operation but adjust by masking out patches from previously segmented instances in the affinity matrix: $W_{ij}^t = \frac{\left(K_i \sum_{s=1}^{t} M_{ij}^s\right)\left(K_j \sum_{s=1}^{t} M_{ij}^s\right)}{\|K_i\|_2 \|K_j\|_2}$ Subsequently, CutLER's learning stage trains a segmentation/detection model with drop-loss, which encourages the model to explore areas not previously identified by MaskCut. An iterative self-training phase is employed for continuously refining the model's performance.

## A.3  Preliminary: Segment Anything Model (SAM) and SA-1B

Inspired by achievement in the NLP field, the Segment Anything project [21] introduces the novel *promptable segmentation task*. At its core lies the Segment Anything Model (SAM) [21], which is capable of producing segmentation masks given user-provided text, points, boxes, and masks in a zero-shot manner. SAM comprises three key components: an MAE [17] pre-trained Vision Transformer [14] that extracts image embeddings, the prompt encoders that embed various types of prompts, and a lightweight Transformer [36] decoder that predicts segmentation masks by integrating image and prompt embeddings.

One significant contribution of SAM [21] is the release of the SA-1B dataset, which comprises 11 million high-resolution images and 1.1 billion segmentation masks, providing a substantial resource for training and evaluating segmentation models. In particular, annotators interactively used SAM to annotate images, and this newly annotated data was then utilized to iteratively update SAM. This cycle was repeated multiple times to progressively enhance both the model and the dataset.

While SAM [21] significantly accelerates the labeling of segmentation masks, annotating an image still requires approximately 14 seconds per mask. Given that each image contains over 100 masks, this equates to more than 30 minutes per image, posing a substantial cost and making it challenging to scale up the training data effectively.

## A.4   Evaluation Datasets

**COCO** (Common Objects in Context) [24] is a widely utilized object detection and segmentation dataset. It consists of 115,000 labeled training images, 5,000 labeled validation images, and more than 200,000 unlabeled images. Its object segmentation covers 80 categories and is mainly on the instance-level. We evaluate our model on COCO `Val2017` with 5000 validation images without training or fine-tuning on any images from the COCO training set. The metrics we choose are class-agnostic COCO style averaged precision and averaged recall for the whole-image inference task, and MaxIoU and OracleIoU for the promptable segmentation task.

**SA-1B** [21] consists of 11 million high-resolution (1500 on average) images and 1.1 billion segmentation masks, approximately 100 masks per image. All masks are collected in a class-agnostic manner with various subject themes including locations, objects, and scenes. Masks cover a wide range of granularity levels, from large-scale objects to fine-grained details. In the whole-image inference task, we randomly selected 1000 SA-1B images that are not used to generate pseudo labels as the validation set.

**LVIS** (Large Vocabulary Instance Segmentation) [15] has 164,000 images with more than 1,200 categories and more than 2 million high-quality instance-level segmentation masks. It has a long tail distribution that naturally reveals a large number of rare categories. In the whole-image inference task, we evaluate our model using its 5000 validation images in a zero-shot manner.

**EntitySeg** [29] is an open-world, class-agnostic dataset that consists of 33277 images in total. There are on average 18.1 entities per image. More than 80% of its images are of high resolution with at least 1000 pixels for the width. EntitySeg also has more accurate boundary annotations. In the whole-image inference task, we evaluate our model with 1314 low-resolution version images ($800 \times 1300$ on average) in a zero-shot manner.

**PACO** (Parts and Attributes of Common Objects) [30] is a detection dataset that provides 641,000 masks for part-level entities not included in traditional datasets. It covers 75 object categories and 456 object-part categories. In the whole-image inference task, we evaluate our model with 2410 validation images in a zero-shot manner.

**PartImageNet** [16] is a large-scale, high-quality dataset with rich part segmentation annotations on a general set of classes with non-rigid, articulated objects. It includes 158 classes and 24,000 images from ImageNet [13]. In the whole-image inference task, we evaluate our model with 2956 validation images in a zero-shot manner.

**ADE20K** [48] is composed of 25,574 training and 2,000 testing images spanning 365 different scenes. It mainly covers semantic-level segmentation with 150 semantic categories and 707,868 objects from 3,688 categories. In the whole-image inference task, we evaluate our model with 2000 testing images in a zero-shot manner.

## A.5   More Visualizations

We provide more qualitative results of UnSAM and UnSAM+ in a zero-shot manner in Figure A1, and Figure A2.

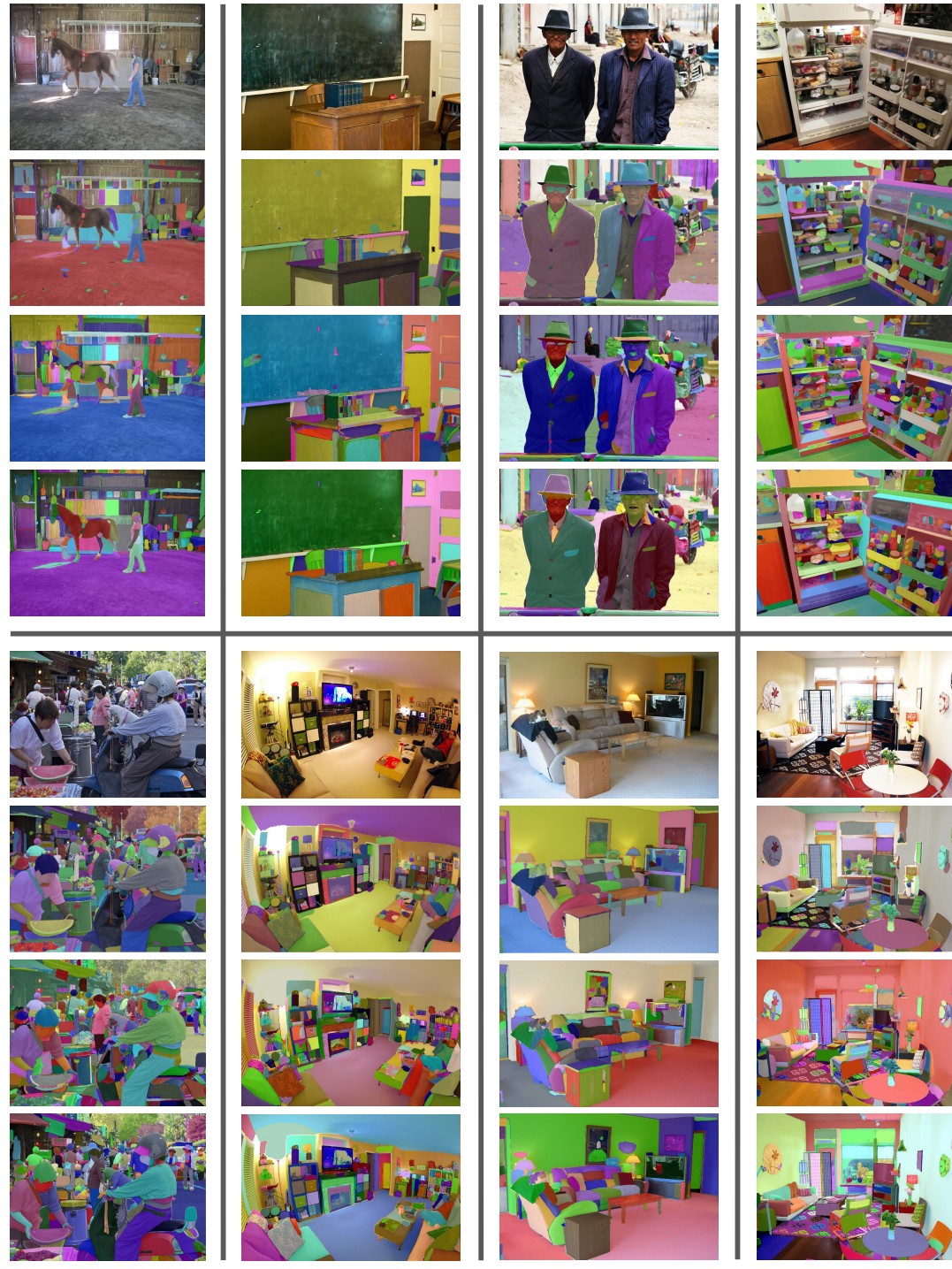

**Figure A1:** More visualizations on COCO [24]. From top to bottom are raw images, segmentation by SAM, segmentation by UnSAM, and segmentation by UnSAM+.

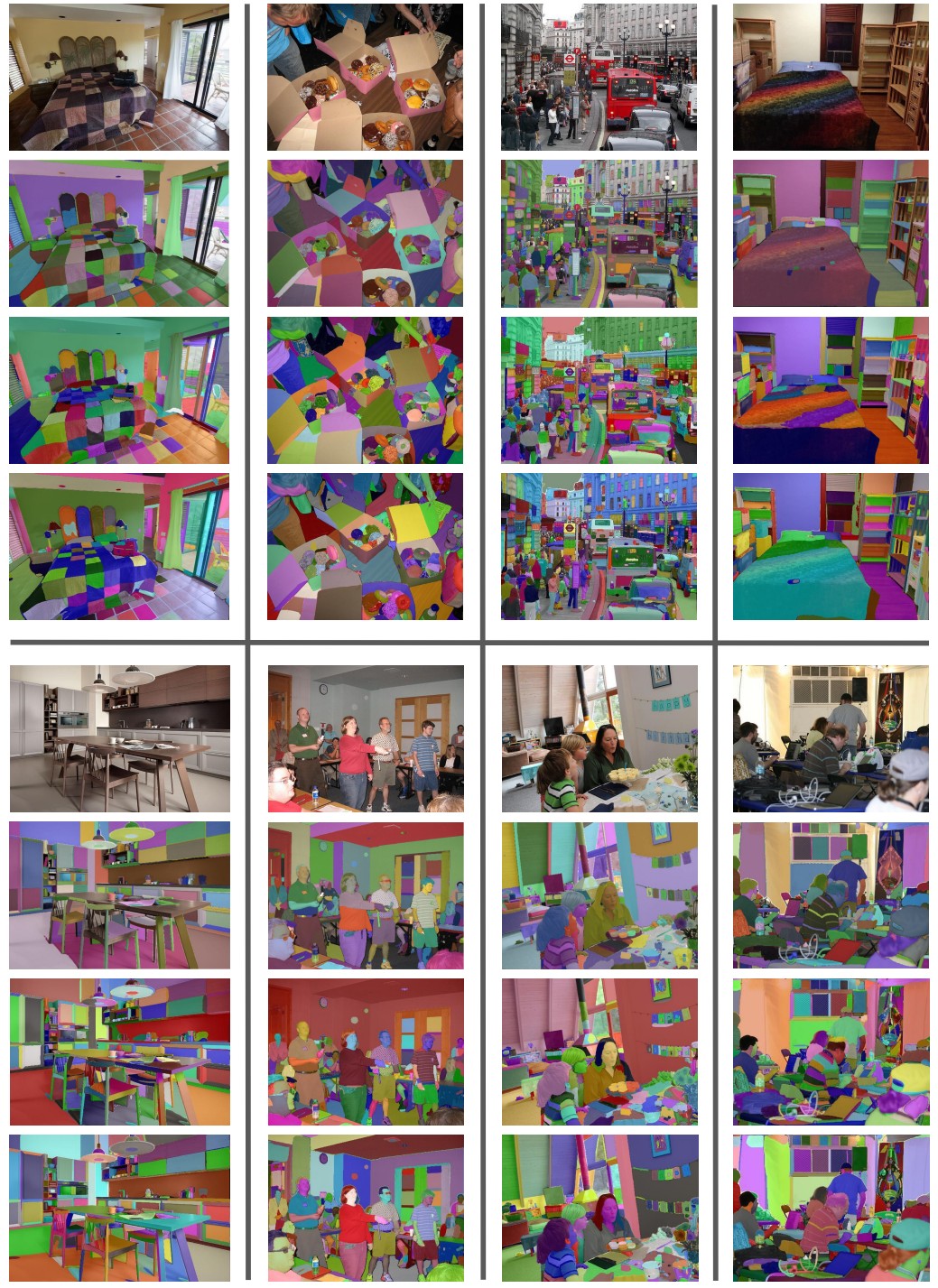

**Figure A2:** More visualizations on PACO [30]. From top to bottom are raw images, segmentation by SAM, segmentation by UnSAM, and segmentation by UnSAM+.

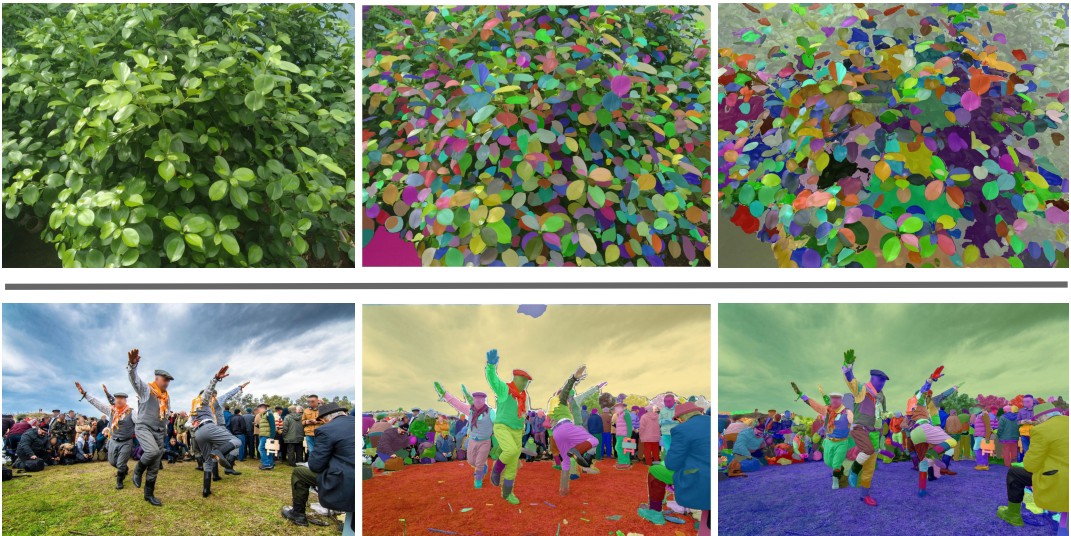

**Figure A3:** Failure cases of UnSAM. From left to right are raw images, segmentation by SAM, and segmentation by UnSAM.

## A.6 Limitations

In images with very dense fine-grained details, UnSAM tends to miss repetitive instances with similar texture. As shown in Figure A3, in the first row, although UnSAM accurately segments the leaves in the center of the picture, it misses some leaves located at the top of the image. Additionally, UnSAM occasionally over-segment images. In the second row, the right sleeve cuff of the dancer has meaningless segmentation masks. This issue mainly arises because the unsupervised clustering method mistakenly considers some information, such as folds and shadows on clothing, as criteria for distinguishing different entities. In contrast, human annotators can use prior knowledge to inform the model that such information should not be valid criteria. In this regard, unsupervised methods still need to close the gap with supervised methods.

## A.7 Ethical Considerations

We train UnSAM and UnSAM+ on ground truths of and pseudo masks generated on SA-1B [21]. SA-1B contains licensed images that are filtered for objectionable content. It is geographically diverse, but some regions and economic groups are underrepresented. Downstream use of UnSAM and UnSAM+ may create their own potential biases.

