# OpenReview forum: "Segment Anything without Supervision"
_NeurIPS.cc/2024/Conference — NeurIPS 2024 poster_

### Official Review · Reviewer_ecwU · 2024-06-24

**Soundness:** 2
**Presentation:** 4
**Contribution:** 2
**Rating:** 6
**Confidence:** 5

**Summary:**

This paper presents Unsupervised SAM (UnSAM) for interactive and automatic whole-image segmentation which does not require human annotations.
This method uses top-down clustering and bottom-up merging to obtain multi-granularity pseudo labels for supervised SAM training. This unsupervised training of SAM achieved good performance on specific datasets.
In addition, this paper finds that SAM can achieve better performance by combining pseudo labels and a small amount of GT from SA-1B for model training.

**Strengths:**

1. The results of this paper are solid.
2. The improvement of the paper on SAM is significant, both in terms of quantitative and partial qualitative results provided.

**Weaknesses:**

1. The motivation of this work is to extend SAM, but it has not been experimentally proven that unsupervised training of UnSAM outperforms fully supervised SAM by continuously increasing the size of the dataset. Instead, it only provides better semi-supervised results.
2. The method used in this paper is very similar to the one published a year ago in [1], which first proposed unsupervised interactive segmentation using top-down clustering and bottom-up merging to obtain hierarchical masks for training interactive segmentation models. I believe that the author needs to provide a clear difference in design compared to [1], rather than just details or differences in model structure and source data.
3. The paper lacks tests of interactive segmentation performance, such as evaluation of NoC (number of clicks) metrics, and should provide a comparison with previous interactive segmentation methods, such as SimpleClick [2] and its subsequent improvement work, etc.

[1] Li, Kehan, et al. "Multi-granularity interaction simulation for unsupervised interactive segmentation." Proceedings of the IEEE/CVF International Conference on Computer Vision. 2023.

[2] Liu, Qin, et al. "Simpleclick: Interactive image segmentation with simple vision transformers." Proceedings of the IEEE/CVF International Conference on Computer Vision. 2023.

**Questions:**

See weaknesses for details. If the author can answer the above questions positively, especially question 2, I will consider raising the score.

**Limitations:**

The problems mentioned in the paper do exist and are difficult to solve, and unsupervised pseudo noise is difficult to avoid.

---

> ### Author Rebuttal · Authors · 2024-08-06
>
> Dear Reviewer ecwU, thank you for your insightful comments, and we really appreciate that you are willing to increase our score if we can answer the questions positively.
> We will provide detailed responses to each of them below.
>
> **[W1] It Has Not Been Experimentally Proven that Unsupervised Training of UnSAM Outperforms Fully Supervised SAM by Continuously Increasing the Size of the Dataset.**
>
> Thanks for your question. **We respectfully argues that surpassing the heavily supervised segmentation model SAM with an unsupervised model like UnSAM is not trivial.** We are pleased to report that UnSAM's performance improves with an increase in training samples—from 0.1% of SA-1B to 0.4% of SA-1B. We found that by increasing the training samples to 0.4% and using a larger backbone, UnSAM's performance already surpasses that of SAM.
>
> | Methods        | Setting      |    # imgs |  Avg. | COCO | LVIS | ADE | Entity | SA-1B | Part-IN | PACO  |
> |:--------       | :--------   | :--------  | --------:| --------:|  --------:| --------:| --------:|  --------:| --------:| --------:|
> | *SAM*     |  *Supervised*    | *11M*  | *42.1* | *49.6* | *46.1* | *45.8* | *45.9* | *60.8* | *28.3* | *18.1*
> | Prev. UnSup. SOTA     |  Unsupervised | 0.2M | 30.1 | 30.5 | 29.1 | 31.1 | 33.5 | 33.3 | 36.0 | 17.1
> | **UnSAM (RN-50)**          |  Unsupervised | 0.1M | 39.2 | 40.5 | 37.7 | 35.7 | 39.6 | 41.9 | 51.6 | 27.5
> | **UnSAM (RN-50)**          |  Unsupervised | 0.4M | 41.1 | 42.0 | 40.5 | **37.5** | 41.0 | 44.5 | 52.7 | 29.7
> | **UnSAM (ViT-B)**          |  Unsupervised | 0.4M | **43.0** | **44.0** | **42.7** | 37.2 | **44.4** | **47.2** | **55.1** | **31.1**
> | *vs. Prev. SOTA*          |  - | - | *+12.9* | *+13.5* | *+13.6* | *+6.1* | *+10.9* | *+13.9* | *+19.1* | *+14.0*
>
> Due to limitations in computing resources, completing the model training on the full 100% SA-1B dataset could take over two months, which may not be feasible within a short rebuttal period. However, the current results already show that UnSAM performs better than SAM on average across seven datasets. We are confident that these performance gains over SAM will further increase as we continue training with more samples.
>
>
> **[W2] Difference in Design Compared to MIS**
>
> - High-level Main Differences in Methods: Unlike MIS, which employs top-down clustering primarily for selecting proposals *WITHOUT* the capability to discover new objects missed by bottom-up clustering methods. In contrast, both bottom-up and top-down clustering in our divide-and-conquer strategy contributes to discovering new objects and parts within an image. This leads to two significant distinctions: 1) Bottom-up clustering relies heavily on low-level features like color and texture, resulting in lower-quality pseudo-masks for instance and semantic segmentation, particularly when dealing with disconnected or overlapping instances. 2) The bottom-up clustering in MIS merges only adjacent groups, often resulting in incomplete segmentation of instances with disconnected parts.
> - Minor Differences in Bottom-up Clustering: While MIS utilizes a cut-based method for bi-partitioning the affinity matrix, our approach employs mean-shift clustering, which leverages cosine similarity for merging groups into larger entities. This makes our method more resilient to noise and outliers in the training data.
> - Model Performance: To ensure a fair comparison, we trained UnSAM using the same model as MIS, ViT-Base, and present the NoC@85 results below:
>
> | Methods        | GrabCut (NoC@85 &#8595;) | Berkeley (NoC@85 &#8595;) | SBD (NoC@85 &#8595;) | DAVIS (NoC@85 &#8595;) |
> |:-------- | --------:| --------:| --------:| --------:|
> | MIS (unsupervised)           | 1.94 | 3.09 | 6.91 | 6.33 |
> | **UnSAM** (unsupervised)     | **1.39** | **1.42** | **3.04** | **4.02** |
> | SimpleClick (supervised)     | 1.40 | 1.44 | 3.28 | 4.10 |
> | **UnSAM+** (semi-supervised) | **1.30** | **1.37** | **2.74** | **3.11** |
>
> The results for MIS and SimpleClick are copied directly from MIS. Our unsupervised model UnSAM significantly surpasses the previous state-of-the-art in unsupervised interactive segmentation and delivers performance comparable to the supervised SimpleClick. And the semi-supervised model UnSAM+ greatly surpasses the SimpleClick model.
>
> It's important to note that datasets such as GrabCut, Berkeley, SBD, and DAVIS are easier and primarily focus on instance and semantic-level masks, typically labeling only a few dominant instances in each image. Consequently, we observe larger performance improvements on the MSCOCO dataset, as detailed in the following table.
>
> **[W3] Comparisons with SimpleClick**
>
> Thank you for introducing us to SimpleClick! Beyond the results shared earlier, we have also measured the 1-IoU for SimpleClick on the MSCOCO dataset and present the results below:
>
> | Methods    | Setting           | 1-IoU  |
> |------------|-------------------|--------|
> | SimpleClick| Supervised        | 52.3   |
> | SAM        | Supervised        | 68.2   |
> | **UnSAM**  | **Unsupervised**  | **59.5** |
> | **UnSAM+** | **Semi-supervised** | **69.5** |
>
> Our unsupervised model, UnSAM, outperforms the supervised SimpleClick by over 7%, and these gains increase to 17.2% under a semi-supervised setting. The more significant performance improvement on MSCOCO, compared to datasets like GrabCut, Berkeley, SBD, and DAVIS, can be attributed to MSCOCO's complexity with numerous small and heavily overlapped instances, which presents a greater challenge than the datasets frequently used for interactive segmentation studies.
>
> *Hope our explanation and experiments can address your inquiries. We will integrate all your valuable comments into our revision!*

---

> > ### Comment · Reviewer_ecwU · 2024-08-08
> >
> > Thank you for the response from the authors. I agree with the performance of unSAM. However, I still have some doubts about the novelty of the method, i.e. **W2**. MIS uses the bottom-up merging strategy to generate pseudo-labels. I think that it is able to merge features of non-adjacent groups according to SSE Cost in MIS. The top-down sampling in MIS is to balance the sampling probability of multi-granularity masks during training, which is indeed different from unSAM.
> >
> > So I'll clarify my question again. I think unSAM's bottom-up clustering is very similar to MIS's bottom-up merging, except that unSAM further subdivides the instance masks produced by top-down clustering. In fact, the result of bottom-up merging contains masks of various granularities (including instance-level masks), so it seems that top-down clustering is not necessary.
> >
> > But I would still consider raising my score, and if the author has time, could you provide some information on how much unSAM performance would be degraded by using only MIS bottom-up merging to generate the mask?

---

> > > ### Author Response · Authors · 2024-08-08
> > > **Thank you for considering to raise the score!**
> > >
> > > Dear Reviewer ecwU,
> > >
> > > We are thrilled to hear that you are considering raising the score!
> > >
> > > Regarding your additional questions about the potential degradation in UnSAM's performance when using only a bottom-up merging approach to generate masks, we have conducted experimental evaluations. We assessed the quality of pseudo-masks generated solely through a bottom-up clustering method on 1,000 images from SA-1B. The results are presented in the table below:
> > >
> > > | Method                        | AR   | AR$_S$ | AR$_M$ | AR$_L$ |
> > > |-------------------------------|------|--------|--------|--------|
> > > | Bottom-up Clustering          | 16.5 | 5.7    | 16.1   | 23.0   |
> > > | Bottom-up + Top-down Clustering | 23.9 | 7.9    | 22.4   | 34.0   |
> > >
> > > The result clearly illustrates that incorporating both bottom-up and top-down clustering methods in our divide-and-conquer strategy significantly enhances performance compared to using bottom-up clustering alone. Notably, the most substantial gains are observed in AR$_L$, underscoring our discussion in the rebuttal that top-down clustering often identifies more instance/semantic-level masks compared to bottom-up methods alone. Unfortunately, due to the limited time for the discussion period, we were unable to complete pseudo-mask generation and segmentation model training for all training samples. However, we anticipate that the performance gains post-model training will align closely with the quality of the pseudo-masks.
> > >
> > > Thank you again for your feedback! Please let us know if there are any more questions. We hope you have a wonderful day!
> > >
> > > Best regards,
> > >
> > > UnSAM Authors

---

> > > > ### Comment · Reviewer_ecwU · 2024-08-09
> > > >
> > > > My main concerns have been addressed and I will raise my rating from 4 to 6. I am very grateful to the authors for their serious approach to the paper and the rebuttal.

---

> > > > > ### Author Response · Authors · 2024-08-10
> > > > >
> > > > > Dear Reviewer ecwU,
> > > > >
> > > > > Thank you so much for raising our score! Hope you have a great weekend!
> > > > >
> > > > > Best,
> > > > >
> > > > > UnSAM Authors

---

### Official Review · Reviewer_wivK · 2024-06-28

**Soundness:** 3
**Presentation:** 2
**Contribution:** 2
**Rating:** 4
**Confidence:** 5

**Summary:**

The paper presents Unsupervised Segment Anything Model (UnSAM) for image segmentation whose training does not have access to human annotations. UnSAM employs a divide-and-conquer approach to hierarchically segment the image. In the divide stage, it uses CutLER [39] to obtain masks, and in the conquer stage, it applies an iterative merging method from SHOES [6]. The proposed method achieves SOTA results.

**Strengths:**

This proposed approach achieves good image segmentation results.

**Weaknesses:**

1. Unclear technical contributions -- limited novelty: The paper uses existing work, including CutLER, SHOES, DINO and Mask2former. The key differences are not explained (well), beyond that these existing methods are put together for unsupervised image segmentation.

2. The performance gain seems to come from additional thresholds used (Lines 130, 142) and potentially unfair comparisons (please see the point 3 and the Questions section below). The thresholds are set in an ad hoc manner.

3. Backbone comparison: In Section 4 (UnSAM), Line 139, the paper mentions using the DINO pre-trained ViT-B/8 encoder, yet none of the tables show results with the ViT-B backbone. Recent methods (e.g., SOHES and SAM) use the ViT-B/8 backbone. A fair comparison with these methods should use the same backbone. Although one might argue that RN-50 and Swin-Tiny (used in this paper) are lighter backbones, using ViT-B/8 would provide a fair comparison.

4. Clarity can be improved:
   - 4.1. Inconsistent Figure Caption: In Figure 1, the caption and figure are not consistent. The caption states “UnSAM (row 2) and SAM (row 3)”, but the figure shows SAM in row 2 and UnSAM in row 3. It is unclear which result corresponds to which method.
   - 4.2. Key Distinctions: The second part of “Key distinctions over prior works on pseudo-mask generation,” Line 167, is unclear. The paper summarizes SOHES [6] but does not explain novelty of the proposed work.

5. Additional comparisons (optional): The paper could be improved by including another comparison with "Unsupervised Universal Image Segmentation," CVPR 24 (published after the NeurIPS deadline, though).

**Questions:**

1. The original CutLER uses ImageNet for unlabeled training, while UnSAM uses the SA-1B dataset. Since the proposed method and baseline use different training data, it is unclear if the proposed method is truly better. Similarly, for the comparison with SOHES on the 0.2M data, did both methods use the same set of 0.2M images?

2. If different sets of 2% unlabeled training data from SA-1B are used, are the results stable? Are the threshold hyperparameters in UnSAM robust to variations of the selected training data?

3. What does “the two extreme cases” in Line 169 refer to?

**Limitations:**

The paper lacks a limitation section. Including a discussion on failure cases would improve the paper.

---

> ### Author Rebuttal · Authors · 2024-08-06
>
> Dear Reviewer wivK, we appreciate your invaluable insights and thoughtful comments. In the following sections, we address the questions you have raised:
>
> **[W1.1] Technical Contributions**
>
> Please check our answers in the global rebuttal. Thank you!
>
> **[W1.2] Differences with Prior Works**
>
> We explained the key differences between UnSAM and prior works (e.g. CutLER and SOHES) in lines 162-173, and described the distinctions in the global rebuttal. Thank you!
>
> **[W2] Performance Gain Comes From Additional Thresholds Used?**
>
> Since SOHES didn't release its codes, models and data, we reproduced SOHES's results and reported the results of SOHES using exactly the same threshold settings as below:
>
> | Pseudo-Labeling   | Threshold Setting  |  AR       | AR$_S$       |  AR$_M$       |  AR$_L$       |
> |:-------- |:--------          | --------:  | --------: | --------: | --------: |
> | SOHES (reported)     |  Official   | 16.4  | 6.0 | 15.8 | 22.6 |
> | SOHES (reproduced)   |  Official   | 16.5  | 5.7 | 16.1 | 23.0 |
> | SOHES (reproduced)   |  Ours       | 16.8  | 6.1 | 16.2 | 22.9 |
> | UnSAM                        |  Ours       | **23.9** | **7.9** | **22.4** | **34.0** |
>
> The performance gain from using additional thresholds is marginal, at only about 0.3%. These results indicate that it is our divide-and-conquer strategy, not the threshold settings, that contributes to the significant performance improvements over SOHES.
>
> **[W3.1] UnSAM Uses a Smaller Backbone (RN-50 / Swin-Tiny) and May be Unfair**
>
> The results for UnSAM reported in the paper were obtained using a significantly smaller backbone and fewer training samples than those used for SAM, placing our method at a disadvantage. To fully address your question, we conducted additional experiments using a larger backbone, ViT-Base, and have presented the comparative results on SA-1B below:
>
> | Method   | Backbone    | Training Data    | AP       | AR      |
> |:-------- |:--------:   | --------:        | --------:|--------:|
> | SAM      |  ViT-Base   |     11M SA-1B    | 38.9     | 60.8
> | UnSAM+   |  RN50       |     0.1M SA-1B   | 42.8     | 64.8
> | UnSAM+   |  ViT-Base   |     0.1M SA-1B   | **44.6**     | **67.2**
>
> UnSAM demonstrates superior performance with a larger backbone, outperforming SAM in terms of both Average Precision (AP) and Average Recall (AR), despite being trained on significantly fewer samples.
>
> **[W3.2] ViT-Base in Line 139** is a typo. It will be addressed in the revision, thank you for pointing it out!
>
> **[W4.1] Inconsistent Figure Caption**: Thank you for pointing this out! We will correct the figure caption accordingly. UnSAM's results are displayed in row 3, while SAM's results are in row 2.
>
> **[W4.1] Clarification on Key Distinctions with SOHES**
>
> Compared to SOHES, UnSAM has 1) better segmentation quality for coarse-grained instance/semantic-level masks. 2) better segmentation quality for fine-grained sub-part masks. For detailed explanations of these improvements, please refer to our responses in the global rebuttal.
>
> **[W5] Additional Comparisons with U2Seg**
>
> Nice question! We have included the comparisons with U2Seg as below:
>
> | Methods   | Venue      |  Backbone |  MSCOCO     |   SA-1B  |    Part-ImageNet  |
> |:--------              |:--------:  | --------:   | --------:| --------:|  --------:|
> | CutLER                |  CVPR 2023 |  RN-50      |    28.1  | 17.0 | 28.7
> | U2Seg                 |  CVPR 2024 |  RN-50      |    27.5  | 19.3 | 29.1
> | SOHES                 |  ICLR 2024 |  ViT-Base   |    30.5  | 33.3 | 36.0
> | UnSAM             |  Ours      |  RN-50      |    **42.0**  | **44.5** | **52.7**
>
> UnSAM outperforms CutLER, U2Seg and SOHES by a large margin on all experimented benchmarks.
>
> **[Q1 Q2] Did UnSAM and SOHES Use the Same Set of 0.2M Images? If Different Sets of 2% Unlabeled Training Data From SA-1B Are Used, Are The Results Stable?***
>
> Unfortunately, SOHES has not released their codes, models, and training data, preventing us from using the identical set of 0.2M images. However, our results remain robust when training the model with different subsets of 0.2M images.
>
> | Methods        |  Seed |  MSCOCO     |   SA-1B  |    Part-ImageNet  |
> |:--------       | --------:   | --------:| --------:|  --------:|
> | Prev. SOTA     |  -    |    30.5  | 33.3 | 36.0
> | UnSAM          |  1    |    41.2  | 43.6 | 52.1
> | UnSAM          |  2    |    42.4  | 44.5 | 52.7
> | UnSAM          |  3    |    41.7  | 44.0 | 51.8
>
> As shown in the table, we observed that the model's performance remains stable across different sets of 2% unlabeled training data (sampled with 3 different seeds). All three models, each trained with a distinct set, consistently outperformed previous state-of-the-art methods by a large margin.
>
> **[Q3] What Does “the two extreme cases” in Line 169 Refer To?**
>
> The term "two extreme cases" refers to the semantic/instance-level masks (the coarsest granularity), and the subpart-level masks (the finest granularity). We will clarify it in the revision.
>
> **[L1] Limitation Section**
>
> We discussed UnSAM's limitations in Section A6 of our submission and will highlight this more prominently in the main paper. Specifically, UnSAM struggles with images containing dense, fine-grained details, often missing repetitive instances with similar textures. Additionally, it tends to over-segment images due to the unsupervised clustering method mistaking details like folds and shadows on clothing as distinct entities—a contrast to human annotators who use prior knowledge to disregard such information. This underscores the ongoing challenge for unsupervised methods to match the performance of supervised approaches.
>
> *Hope our explanation and experiments can address your inquiries. We will integrate all your valuable comments into our revision!*

---

> > ### Comment · Reviewer_wivK · 2024-08-12
> > **The rebuttal successfully addressed my comments**
> >
> > The authors' response successfully addressed my concerns, and I would like to increase the paper's rating to "weak accept".

---

> > > ### Author Response · Authors · 2024-08-12
> > >
> > > Dear Reviewer wivk,
> > >
> > > Thank you so much for raising our score from 4 to 6! Hope you have a wonderful day!
> > >
> > > Best,
> > >
> > > UnSAM Authors

---

> > ### Author Response · Authors · 2024-08-12
> >
> > Dear Reviewer wivK,
> >
> > Thank you for your thorough review and insightful feedback on our paper. Your input has been instrumental in refining the quality of our work.
> >
> > We are pleased to report that we have carefully addressed each of the concerns you raised. Here's a brief summary of our actions:
> >
> > - **Technical Contributions**: We have detailed our key technical contributions in the global rebuttal and provided quantitative results that underscore the significant performance improvements these contributions bring.
> >
> > - **Ablation Study on Thresholds and Backbones**: In response to your comments, we conducted further experiments with various thresholds and backbones. The results, presented in the tables at W2 and W3.1, indicate that a larger backbone significantly enhances our outcomes.
> >
> > - **Comparison with U2Seg**: We have included a comparison with U2Seg in section W5. The results demonstrate that UnSAM significantly outperforms previous methods in unsupervised image segmentation.
> >
> > - **Robustness Across SA-1B Subsets**: We have tested our method across various subsets of SA-1B, consistently achieving over 10% performance gains, which validates the robustness of our approach.
> >
> > We remain open and committed to addressing any additional questions or concerns. Your feedback continues to shape our research, and we appreciate your contribution to the improvement of our paper.
> >
> > Please feel free to reach out with further questions or suggestions. We look forward to potentially discussing these matters in greater depth and continuing to refine our work with your valued expertise.
> >
> > Warm regards,
> >
> > UnSAM Authors

---

### Official Review · Reviewer_p86w · 2024-07-10

**Soundness:** 2
**Presentation:** 4
**Contribution:** 3
**Rating:** 6
**Confidence:** 4

**Summary:**

The paper explores a new way to generate hierarchical pseudo-masks to train downstream segmentation models without human annotations. First, the image is segmented using CutLER. Then, within CutLER proposed masks (with cropping and resizing), DINO features are extracted, and patches are iteratively merged based on proximity in cosine distance. The masks are then refined and filtered using CRF/CascadePSP. The pool of pseudo-labels is used to train Mask2Former architecture or Semantic-SAM architecture, for whole-image or prompt-able segmentation tasks, respectively. Additional self-training is performed to improve results. The evaluation is carried out on several datasets using the average recall (AR) metric. UnSAM shows significant improvement over other models in terms of recall of generated masks and final outputs.

**Strengths:**

1) The method presented in the paper shows improved results in unsupervised segmentation. Reducing the recall gap between supervised SAM and unsupervised methods.
2) The additional masks generated using the unsupervised approach in combination with manual annotatons to enhance the performance further, surpassing that of SAM.

**Weaknesses:**

#### What is "UnSAM"?
The naming scheme adopted in the paper is slightly confusing. UnSAM refers to all:
 - the hierarchical pseudo-labelling scheme building on top of CutLER,
 - the distillation of such scheme to Mask2Former architecture,
 - distillation of such a scheme into semantic-SAM architecture.

While it still is possible to parse the results in the tables based on the setting, the ability to understand and scrutinise the text is severely impacted. Note the distinction is important as the output of three of these appears to be different.

#### Is AR a sensible metric?
The proposed model seems to generate an extremely large amount of masks (up to 2000, L407). While it seems that some useful masks are in this set, the cardinality of the output makes one wonder if it is useful. What is the number of masks used for assessing recall? Is it 1000? Cutler has a setting of 100-200 from published configs. Clearly, increasing the number of masks boosts recall. It is important that the evaluation maintains the same amount of masks for all methods.

It also raises a question whether some classical hierarchical segmentation methods such as MCG [A] or Arbelaez et al. [B] would get similar or close performance when such large pool of candidates is allowed.

#### What is the effective difference between a proposal and SOHES?
The hierarchical merging scheme follows SOHES formulation, with the difference being the CutLER region "proposal" as a constraint. Is that all? It is important to highlight the differences here to both figure out the novelty and correctly attribute improvements in performance to different components/steps of the proposal. Additionally, it appears that distillation to Mask2Former abandons the hierarchical association. Does this mean that the output is no longer organised in a hierarchical manner?

#### What makes the method work better than prior works?
There is some lack of ablations that explore the influence and sensitivity of various components in the pipeline. The paper only reports on the construction and technical details of the UnSAM methods and the evaluation of this approach. It lacks analysis and insights into what the makes the construction effective. The advancement of knowledge offered by the paper is somewhat limited.

#### Are different two different models required?
It is not entirely clear why the two modes of operation, "whole image" and "prompt-able" require different models/architectures. Could the Semantic-SAM-based model be prompted in a similar fashion to SAM (i.e. a grid of points) to perform the whole image segmentation? Is this disadvantageos for the whole-image segmentation task?

#### Inaccuracies
Finally, it is important to note that CascadePSP relies on manual annotations. Thus, unSAM + CascadePSP is _slightly_ supervised.

Since UnSAM partly distils CutLER outputs, would it not be more accurate to incorporate the #images used in CutLER to those in UnSAM, writing e.g. 1.4M in Table 1 instead of 0.1M?

[Nit] L135: "<...> we employ iterative _merging_ to _decompose_" might require rephrasing as it is currently a slight oxymoron.

---

[A] Arbelaez et al. "Multiscale combinatorial grouping"

[B] Arbelaez et al. "Contour detection and hierarchical image segmentation"

**Questions:**

The central questions to address in the rebuttal would be around the evaluation protocol. Some more explanation about the contribution of the paper would also be helpful.


While the paper presents strong results, the appropriateness of the evaluation protocol is somewhat questionable. Furthermore, there are some questions around the difference of the proposed scheme to prior work, and lack of experiments and analysis to explain it. While a rebuttal can address these issues, I currently rate the paper as a Borderline Reject.

**Limitations:**

Limitation are appropriately addressed.

---

> ### Author Rebuttal · Authors · 2024-08-06
>
> Dear Reviewer p86w, thank you for your thoughtful comments. We will provide detailed responses to your questions below:
>
> **[W1] What is "UnSAM"?**
>
> Nice suggestion. In the paper, UnSAM stands for **Un**supervised **S**egment **A**nything **M**odel. The pseudo-labeling strategy discussed in our paper is named as divide-and-conquer. I agree that we can make this clearer, and we plan to designate the terms UnSAM (pseudo), UnSAM (whole-image), and UnSAM (promptable) to distinctly refer to the pseudo-labeling method, whole-image segmentation model, and promptable image segmentation models, respectively. We would appreciate any suggestions you might have!
>
> **[W2] Why Using AR as One of The Main Evaluation Metrics?**
>
> Excellent question. We explained the rationale for using AR as a primary evaluation metric in lines 234-238, following previous works like CutLER (CVPR2023) and SOHES (ICLR2024). But **why is AR more appropriate than AP for unsupervised segmentation?** The main reason is that all human-labeled datasets, including SA-1B, only label a subset of objects, and AR doesn't penalize models for detecting objects not labeled in these datasets.
>
> Below is a comparison of results on the MSCOCO dataset that further illustrates why AR is preferred:
>
> | Method   | Venue  |  Setting |Backbone | Training Data    | AP       | AR       |
> |:-------- |:--------   |:-------- | --------:        | --------:| --------:| --------:|
> | SAM      |  ICCV 2023   |  supervised     | ViT-Base   |     11M SA-1B    | 5.9      | 49.4     |
> | CutLER  |  CVPR 2023 |  unsupervised | RN-50       |     1.2M ImageNet    | 6.5      | 31.4     |
> | SOHES  |  ICLR 2024   |  unsupervised | ViT-Base   |     0.2M SA-1B   | 2.0      | 30.5     |
> | **UnSAM**   |  Ours      |  unsupervised | RN-50      |     0.4M SA-1B   | 3.6      | 42.0     |
> | **UnSAM+**   |  Ours    |  lightly supervised | RN-50      |     0.1M SA-1B   | **7.1**      | **52.3**     |
>
> Despite SAM's overall better performance, its lower AP compared to CutLER highlights that AP can unfairly penalize models for detecting unannotated objects in evaluation datasets, failing to truly reflect a model’s capabilities. I agree that AR is not without its imperfections, but it is a more suitable metric than AP for unsupervised segmentation. Future research on developing new metrics is needed.
>
> **[W3.1] The Proposed Model Seems to Generate an Extremely Large Amount of Masks.**
>
> Sorry for the confusion—the 2000 queries mentioned in L407 refer to the number of learnable queries in Mask2Former, not the number of masks predicted. Before outputting the final model predictions, we apply non-maximum suppression, confidence score thresholding, etc. Consequently, the average number of masks our model predicts is around 500.
>
> **[W3.2] What is The Number of Masks Used for Assessing Recall? Why Using AR$_{1000}$?**
>
> We assessed recall using AR$_{1000}$, following the approach used by SOHES. The maximum number of masks per image is set at 1000 for UnSAM and CutLER, and 32*32 (=1024) for SAM. We set the maximum number of masks as 1000 in CutLER's config.
>
> **[W4 W5] What's the Difference Between UnSAM and SOHES? Why UnSAM Performs Better Than Prior Works?**
>
> Good question! We have outlined the key differences between UnSAM and prior works (e.g. CutLER and SOHES) in lines 162-173. These distinctions are further detailed in the global rebuttal. Thank you for checking!
>
> **[W6] Are Different Two Different Models Required?**
>
> It is indeed feasible to use the same model for both tasks. However, we choose to employ two distinct models primarily due to differences in inference time. While the Semantic-SAM-based model can be prompted similarly to SAM (using a grid of points) with only a minor performance disparity (< 2~3%), its processing speed is significantly slower—at least 5 times slower than a specialized whole-image segmentation framework like Mask2Former. In future research, we plan to utilize models such as FastSAM or SAM-2 to enhance the speed of interactive segmentation models.
>
> Additionally, we can still effectively construct the hierarchical structure using the masks from Mask2Former by post-processing the results based on mask overlaps.
>
> **[W7] CascadePSP**
>
> Great question! We follow previous works CutLER (CVPR 2023) and SOHES (ICLR 2024), and utilize CascadePSP (from SOHES) or CRF (from CutLER) for mask refinement (Table 3). We opted for CascadePSP as our default method primarily for two reasons:
>
> 1) **Efficiency**: The primary advantage of CascadePSP is not actually in terms of model performance, but rather in terms of inference speed. In our local small scale experiments, we discovered that the performance difference (in terms of AR) between training UnSAM with CascadePSP-refined pseudo-masks and training UnSAM with CRF-refined masks is only about 2-3%, but the speed difference in refining pseudo-masks is about 5-10 times. Due to our limited computing resources, refining pseudo-masks using CRF would take approximately 3-4 months, making it unfeasible.
>
> 2) **Consistency with Previous Works**: To maintain consistency with SOHES, we used the same refinement method.
>
> To fully answer your question, how can we speed up the overall process for CRF-based mask refinement? One strategy could be to train a CascadePSP model using the "ground-truth" generated by CRF to achieve "fully-unsupervised" mask refinement quickly.
>
> **[W8] #images Should Include ImageNet**
>
> Since ImageNet was actually used by all methods for pre-training the backbone, including SAM, SOHES and CutLER, and UnSAM was never trained on pseudo-labels from ImageNet, we didn't include it as training data. We'll explain it in our paper.
>
> **[W9] Typos and Minors Issues** will be addressed in the revision! Thank you for pointing them out!
>
> *Hope our explanation and experiments can address your questions. We will integrate your valuable comments into our revision!*

---

> > ### Comment · Reviewer_p86w · 2024-08-11
> >
> > I thank the authors for their response. I agree with the proposed changes.
> >
> > #### Difference to SOHES
> >
> > I thank the authors for the added explanation highlighting the output difference between SOHES and UnSAM. The question was about the formulation.
> >
> > > SOHES heavily utilizes low-level feature similarities between patches for cluster merging.
> >
> > My current understanding is that UnSAM does as well (L140-142). It seems that this does not create a problem in this case due to the use of CutLER as the overall limit of the highest level in the hierarchy, correct?
> >
> > Other than the use of CutLER to limit these masks, are there any other noteworthy differences in the proposed Divide-and-Conquer?

---

> > > ### Author Response · Authors · 2024-08-12
> > >
> > > Dear Reviewer p86w,
> > >
> > > Thank you for your response. We are pleased that our rebuttal has addressed some of your concerns! Here are our detailed answers to your additional queries:
> > >
> > > **1) SOHES heavily utilizes low-level feature similarities between patches for cluster merging. It seems that this does not create a problem for UnSAM due to the use of CutLER as the overall limit of the highest level in the hierarchy, correct?**
> > >
> > > Yes, the issue of over-relying on low-level similarities is mitigated by our overall divide-and-conquer pipeline. Divide-and-conquer enables our bottom-up clustering to focus on selected regions, performing cluster merging effectively without being influenced by outliers outside these regions that are identified during the divide stage.
> > >
> > > **2) Distinctiveness of Divide-and-Conquer:**
> > >
> > > Beyond the differences listed in our global rebuttal, here we outline additional distinctions:
> > > - **Major Differences in the Overall Pipeline**: *We wish to emphasize that the core contribution of this work lies in the integration of both bottom-up and top-down clustering—termed the Divide-and-Conquer strategy, a simple yet effective change with significant performance gains.* As no previous work has used this pipeline for unsupervised image segmentation, we believe that our overall pipeline is novel and adds new knowledge to the field. With advancements in either top-down or bottom-up clustering methods, we believe the performance of our Divide-and-Conquer approach can be further enhanced. Thus, we assert that the significant and innovative contribution of our work is the overall pipeline itself, rather than the individual components of each stage.
> > >
> > > - **Major Difference Between Our Bottom-up Clustering and SOHES**: The use of the divide phase significantly impacts our bottom-up clustering in two key ways:
> > >    1) **Outlier Removal**: Our bottom-up clustering focuses exclusively on regions identified during the top-down clustering phase, which sharpens the model's focus and effectively filters out many noisy outliers outside these regions that are identified during the divide stage.
> > >    2) **Two Stage Pseudo-labeling**: The instance/semantic-level masks generated by CutLER specify candidate regions within the image. Similar to the use of selective search in R-CNN or the Region Proposal Network (RPN) in Faster/Mask R-CNN, CutLER primarily functions to provide initial candidate regions for subsequent detection or segmentation stages. This two-stage approach allows our bottom-up clustering to zoom in on these defined regions, facilitating the detection of smaller objects that SOHES often misses.
> > >
> > > - **Performance Comparison**: We have reported substantial performance gains over SOHES, particularly in AR$_L$, supporting our argument that top-down clustering is more effective at identifying instance/semantic-level masks compared to solely bottom-up approaches. Due to the lack of publicly available codes and models from SOHES, we compare against our re-implemention of SOHES for fair comparisons.
> > >
> > >     | Method                        | AR    | AR$_S$ | AR$_M$ | AR$_L$ |
> > >     |-------------------------------|-------|--------|--------|--------|
> > >     | SOHES                         | 16.5  | 5.7    | 16.1   | 23.0   |
> > >     | UnSAM                         | **23.9** | **7.9**  | **22.4** | **34.0** |
> > >     | *Improvement over SOHES*      | +7.4  | +2.2   | +6.3   | +11.0  |
> > >
> > > The below table showcases how training segmentation models on high-quality pseudo-masks from our strategy significantly advances the state-of-the-art in unsupervised segmentation across multiple datasets by an average of 12.9%.
> > >
> > > | Methods        |  # imgs |  Avg. | COCO | LVIS | ADE | Entity | SA-1B | Part-IN | PACO |
> > > |:--------       | --------:   | --------:| --------:|  --------:| --------:| --------:|  --------:| --------:| --------:|
> > > | SOHES (ViT-Base)       | 0.2M | 30.1 | 30.5 | 29.1 | 31.1 | 33.5 | 33.3 | 36.0 | 17.1 |
> > > | UnSAM (RN-50)          | 0.1M | 39.2 | 40.5 | 37.7 | 35.7 | 39.6 | 41.9 | 51.6 | 27.5 |
> > > | UnSAM (RN-50)          | 0.2M | 40.4 | 41.2 | 39.7 | 36.8 | 40.3 | 43.6 | 52.1 | 29.1 |
> > > | UnSAM (RN-50)          | 0.4M | 41.1 | 42.0 | 40.5 | 37.5 | 41.0 | 44.5 | 52.7 | 29.7 |
> > > | UnSAM (ViT-B)            | 0.4M | **43.0** | **44.0** | **42.7** | **37.2** | **44.4** | **47.2** | **55.1** | **31.1** |
> > > | *vs. SOHES*               |          | *+12.9* | *+13.5* | *+13.6* | *+6.1* | *+10.9* | *+13.9* | *+19.1* | *+14.0* |
> > >
> > >
> > > **We would like to emphasize our commitment to simple science: we hold the view that straightforward changes leading to significant performance improvements are much more valuable than complex modifications that yield only minimal benefits.**
> > >
> > > Thank you once again for your insights! We welcome any further questions. Have a fantastic day!
> > >
> > > Best regards,
> > >
> > > UnSAM Authors

---

> > > > ### Comment · Reviewer_p86w · 2024-08-13
> > > >
> > > > I thank the authors for their thorough reply. Simple improvements that lead to significant gains are noteworthy and important. I was merely looking to confirm I understood the formulation that led to such improvements correctly. Lines 167-173 give the impression that bottom-up clustering in [6] is formulated differently.
> > > >
> > > > I have updated my recommendation.

---

> > > > > ### Author Response · Authors · 2024-08-13
> > > > >
> > > > > Dear Reviewer p86w,
> > > > >
> > > > > Thank you very much for raising our score! We are pleased that our rebuttal addressed your concerns. We truly appreciate the constructive discussions with you, which can definitely help us refine our paper.
> > > > >
> > > > > Best regards,
> > > > >
> > > > > UnSAM Authors

---

### Official Review · Reviewer_S6W7 · 2024-07-13

**Soundness:** 3
**Presentation:** 3
**Contribution:** 2
**Rating:** 6
**Confidence:** 4

**Summary:**

This paper proposes an approach to generate masks from images in an unsupervised manner, which are then used to train segmentation models. The major distinction from previous works is the proposed divide-and-conquer strategy, which first adopts the CutLER to obtain coarse instance/segmentation masks and then uses iterative merging like the SOHES method to generate fine-grained masks. Experiments are conducted on different datasets with detailed analysis and advanced performance.

**Strengths:**

- The paper is well-written and easy to follow. Many technical details are provided. The reviewer believes the results should be easy to reproduce.
- The two-stage approach that adopts both CutLER and SOHES to generate pseudo masks is reasonable.
- The overall performance surpasses previous SOTA methods.

**Weaknesses:**

- The major concern is about technical novelty. Though effective, the proposed method technically adds a candidate extraction method before the previous method's iterative refinement stage. All these methods already exist in the unsupervised segmentation field. Therefore, although the performance improvement is reasonable, the reviewer thinks its technical novelty is limited as a NeurIPS research paper.
- The comparison with SAM is based on different backbones. Meanwhile, some details, such as the number of masks (UnSAM's 6 vs. SAM's 3), are different. It may have a subtle influence on the performance comparison.
- Some typos exist. For example, the $\theta_{t+1}$ in line-147 should be $\theta_{t-1}$.

**Questions:**

Please refer to the weaknesses for the concerns. The reviewer would appreciate it if these concerns could be addressed after the rebuttal.

**Limitations:**

The authors discussed the limitations in the appendix.

---

> ### Author Rebuttal · Authors · 2024-08-06
>
> Dear Reviewer S6W7, we appreciate your invaluable insights and thoughtful comments! In the following sections, we address the questions you have raised:
>
> **[W1] Contributions and Insights**
> 1. *Novel and Simple Pipeline:* We introduce a simple yet effective divide-and-conquer pipeline for producing high-quality pseudo-masks for unsupervised image segmentation. As no previous work has used this pipeline for unsupervised image segmentation, we believe that our overall pipeline is novel and adds new knowledge to the field. *We believe in simple science: we think a simple change leading to substantial performance gains is far more valuable and intriguing than a complex method yielding marginal gains.*
> 2. UnSAM, for the first time, demonstrates that unsupervised image segmentation method can achieve competitive results with the supervised counterpart SAM. In addition, UnSAM achieves over 12% better AR than the previous unsupervised segmentation methods.
> 3. UnSAM is also the first to show that the state-of-the-art supervised segmentation method, SAM, can benefit from our self-supervised labels—a discovery that has not been previously reported. UnSAM+ exceeds SAM’s AP by 3.9% and AR by over 6.7% on SA-1B.
>
> **[W2.1] Model Backbones**
>
> The results for UnSAM reported in the paper were obtained using a significantly smaller backbone and fewer training samples than those used for SAM, placing our method at a disadvantage. To fully address your question, we conducted additional experiments using a larger backbone, ViT-Base, and have presented the comparative results on SA-1B below:
>
> | Method   | Backbone    | Training Data    | AP       | AR      |
> |:-------- |:--------:   | --------:        | --------:|--------:|
> | SAM      |  ViT-Base   |     11M SA-1B    | 38.9     | 60.8
> | UnSAM+   |  RN50       |     0.1M SA-1B   | **42.8**     | **64.8**
> | UnSAM+   |  ViT-Base   |     0.1M SA-1B   | **44.6**     | **67.2**
>
> UnSAM demonstrates superior performance with a larger backbone, outperforming SAM in terms of both Average Precision (AP) and Average Recall (AR), despite being trained on significantly fewer samples.
>
> **[W2.2] Number of Masks Per Point**
>
> Great question! We employed additional granularity levels (i.e., more masks per click) because our unsupervised pseudo-labeling method can create a hierarchical structure with more granularity levels than the ground-truth masks from SA-1B. We increased the number of masks per click to fully utilize the advantages of our hierarchically structured data. Despite increasing the number of masks for SAM (two clicks producing six output masks), the performance improvement on MSCOCO was relatively marginal.
>
> | Method   | Backbone (# params)    | Training Data    | # Masks  | 1-IoU    |
> |:-------- |:--------:   | --------:        | --------:| --------:|
> | SAM      |  ViT-Base (85M)  |     11M SA-1B    | 3     | 68.2     |
> | SAM      |  ViT-Base (85M)  |     11M SA-1B    | 6     | 69.0     |
> | UnSAM+   |  Swin-Tiny (25M) |     0.1M SA-1B   | 6     | **69.5**     |
> | UnSAM+   |  Swin-Tiny (25M) |     0.4M SA-1B   | 6     | **70.4**     |
>
> As indicated in the table, UnSAM+ surpasses SAM by over 1.4% despite being trained with 100 times fewer samples.
>
> **[W3] Typos**
>
> Thank you for pointing out these typos! We will correct them and thoroughly review the manuscript before finalizing the paper.
>
> *Hope our explanation and experiments can address your inquiries. We will integrate all your valuable comments into our revision!*

---

> > ### Author Response · Authors · 2024-08-12
> >
> > Dear Reviewer S6W7,
> >
> > Thank you for your thorough review and insightful feedback on our paper. Your input has been instrumental in refining the quality of our work.
> >
> > We are pleased to report that we have carefully addressed each of the concerns you raised. Here's a brief summary of our actions:
> >
> > - **Technical Contributions**: We have detailed our key technical contributions in the global rebuttal and provided quantitative results that underscore the significant performance improvements these contributions bring.
> >
> > - **Ablation Study on Backbones**: In response to your comments, we conducted further experiments with various backbones. The results, presented in the tables at W2.1, indicate that a larger backbone significantly enhances our outcomes.
> >
> > - **Ablation Study on Number of Masks**: In response to your comments, we conducted further experiments with various number of masks. The results, presented in the tables at W2.2, indicate that UnSAM can still achieve performance gains over SAM as we increase the number of masks of SAM.
> >
> > We remain open and committed to addressing any additional questions or concerns. Your feedback continues to shape our research, and we appreciate your contribution to the improvement of our paper.
> >
> > Please feel free to reach out with further questions or suggestions. We look forward to potentially discussing these matters in greater depth and continuing to refine our work with your valued expertise.
> >
> > Warm regards,
> >
> > UnSAM Authors

---

> > > ### Comment · Reviewer_S6W7 · 2024-08-13
> > >
> > > Thanks for the response and additional information. It addressed my major concerns. Therefore, I think the recommendation score can be improved to Weak Accept.

---

> > > > ### Author Response · Authors · 2024-08-13
> > > >
> > > > Dear Reviewer S6W7,
> > > >
> > > > Thank you very much for raising our score! We are pleased that our rebuttal addressed your concerns. Hope you have a nice day!
> > > >
> > > > Best regards,
> > > >
> > > > UnSAM Authors

---

### Author Rebuttal · Authors · 2024-08-06

We would like to thank the reviewers for their valuable feedback. In this paper, we present Unsupervised SAM (UnSAM) for promptable and automatic whole-image segmentation that does not require human annotations. We are encouraged by the acknowledgements on:

- **Comprehensive Experiments and SOTA Results**: We are especially glad that the reviewers believe that "the results of this paper are solid"(ecwU). Reviewers also noted that UnSAM "achieves improved results over state-of-the-art (SOTA) methods" (S6W7, wivK, p86w), and that "the improvement of the paper on SAM is significant, both in terms of quantitative and partial qualitative results provided" (ecwU).
- **Novel Setup**: UnSAM not only demonstrates SOTA performance in unsupervised segmentation but also shows that "the additional masks generated using the unsupervised approach, in combination with manual annotations, enhance performance further, surpassing that of SAM" (p86w). And it was agreed that "the two-stage approach to generate pseudo masks is reasonable." (S6W7)

We really appreciate that reviewers ecwU and p86w have indicated they might increase our score based on the content of our rebuttal. We are more than happy to discuss with all reviewers to address any additional questions during the discussion period.

In this section, we commence by tackling the concerns that have been collectively raised.

**Technical Contributions of UnSAM**
- *Novel and Simple Pipeline:* We introduce a simple yet effective divide-and-conquer pipeline for producing high-quality pseudo-masks for unsupervised image segmentation. As no previous work has used this pipeline for unsupervised image segmentation, we believe that our overall pipeline is novel and adds new knowledge to the field.
- UnSAM, for the first time, demonstrates that *unsupervised image segmentation method can achieve competitive results with the supervised counterpart SAM*. As shown in the table below, UnSAM achieves over 12% better AR than the previous unsupervised segmentation methods.
- UnSAM is also *the first to show that the state-of-the-art supervised segmentation method, SAM, can benefit from our self-supervised labels*—a discovery that has not been previously reported. As illustrated in the table below, UnSAM+ surpasses SAM's Average Precision (AP) by 3.9% and Average Recall (AR) by over 6.7% on SA-1B.
- *We believe in simple science*: We think a simple change leading to substantial performance gains is far more valuable and intriguing than a complex method yielding marginal gains.

| Methods        | Setting      |    # imgs |  Avg. | COCO | LVIS | ADE | Entity | SA-1B | Part-ImageNet | PACO  |
|:--------       | :--------   | :--------  | --------:| --------:|  --------:| --------:| --------:|  --------:| --------:| --------:|
| Prev. UnSup. SOTA (ViT-B)     |  Unsupervised | 0.2M | 30.1 | 30.5 | 29.1 | 31.1 | 33.5 | 33.3 | 36.0 | 17.1
| UnSAM (RN-50)          |  Unsupervised | 0.1M | 39.2 | 40.5 | 37.7 | 35.7 | 39.6 | 41.9 | 51.6 | 27.5
| UnSAM (RN-50)          |  Unsupervised | 0.4M | 41.1 | 42.0 | 40.5 | **37.5** | 41.0 | 44.5 | 52.7 | 29.7
| UnSAM (ViT-B)          |  Unsupervised | 0.4M | **43.0** | **44.0** | **42.7** | 37.2 | **44.4** | **47.2** | **55.1** | **31.1**
| *vs. Prev. SOTA*          |  - | - | *+12.9* | *+13.5* | *+13.6* | *+6.1* | *+10.9* | *+13.9* | *+19.1* | *+14.0*

| Methods        | Setting      |    # imgs |  Avg. | COCO | LVIS | ADE | Entity | SA-1B | Part-ImageNet | PACO  |
|:--------       | :--------   | :--------  | --------:| --------:|  --------:| --------:| --------:|  --------:| --------:| --------:|
| SAM     |  Fully-Supervised                                           | 11M  | 42.1      | 49.6      | 46.1      | **45.8** | 45.9       | 60.8      | 28.3 | 18.1
| UnSAM+ (RN-50)          |  Lightly-Supervised | **0.1M** | **48.8** | **52.2** | **50.8** |   45.3    | **49.8**  | **64.8** | **46.0** | **32.3**
| *vs. SAM*          |  - | - | *+6.7* | *+2.6* | *+4.7* | -0.5 | *+3.9* | *+4.0* | *+17.7* | *+14.2*

**What's the Difference Between UnSAM and Prior Works? Why UnSAM Performs Better Than Prior Works?**

- **Comparison with CutLER and U2Seg**: These models are limited to providing only instance/semantic-level masks, missing the hierarchical structure that is often present in complex visual scenes. In contrast, our pipeline captures this hierarchical structure by identifying more fine-grained pixel clusters.

- **Comparison with SOHES**:
  1. Segmentation Quality at Instance/Semantic Level: SOHES heavily utilizes low-level feature similarities between patches for cluster merging, which often leads to missing many instance masks that have disconnected or occluded components. SOHES struggles to recognize that visually distinct patches (e.g., red-colored t-shirts and green-colored pants) may belong to the same instance (e.g., a person). In contrast, our divide-and-conquer strategy employs a cut-based method that evaluates both the total dissimilarity between different pixel groups and the total similarity within these groups during image partitioning. UnSAM often identifies instance/semantic-level masks that SOHES overlooks. Consequently, UnSAM achieves a recall that is **1.5 times higher than SOHES for large objects** on SA-1B.

   2. Segmentation Quality for Part/Sub-part Masks: Our use of top-down clustering also allows the model to zoom in on selected regions provided by our cut-based clustering method, resulting in more detailed and fine-grained masks for small objects. Consequently, UnSAM exhibits a recall rate that is **1.3 times higher than SOHES for small objects** on SA-1B.

Because UnSAM can produce higher quality pseudo-labels than both CutLER and SOHES, the resulting segmentation model trained on these masks can get better model performance.

**We will integrate all the valuable suggestions into our final version and open-source the code.** Next, we address all concerns raised by the reviewers below.

---

### Decision · Program_Chairs · 2024-09-25

**Decision:**

Accept (poster)

**Comment:**

After rebuttal and discussion, all reviewers assess the manuscript with a weak accept (note that in one case this was only written in a comment and the numerical score has not been updated). The main strengths are the good presentation of the proposed method and the high performance. The main weakness is the limited novelty.